# Neural Polynomial Gabor Fields for Macro Motion Analysis

**Chen Geng[1,*], Hong-Xing Yu[1,*], Sida Peng[2], Xiaowei Zhou[2], Jiajun Wu[1]**
[1]Stanford University  [2]Zhejiang University
{gengchen,koven,jiajunwu}@cs.stanford.edu
{pengsida,xwzhou}@zju.edu.cn

## Abstract

We study macro motion analysis, where macro motion refers to the collection of all visually observable motions in a dynamic scene. Traditional filtering-based methods on motion analysis typically focus only on local and tiny motions, yet fail to represent large motions or 3D scenes. Recent dynamic neural representations can faithfully represent motions using correspondences, but they cannot be directly used for motion analysis. In this work, we propose Phase-based neural polynomial Gabor fields (Phase-PGF), which learns to represent scene dynamics with low-dimensional time-varying phases. We theoretically show that Phase-PGF has several properties suitable for macro motion analysis. In our experiments, we collect diverse 2D and 3D dynamic scenes and show that Phase-PGF enables dynamic scene analysis and editing tasks including motion loop detection, motion factorization, motion smoothing, and motion magnification. Project page: https://chen-geng.com/phasepgf

## 1 Introduction

The physical world is dynamic and full of different motions: from waving trees to walking people, motions are present in different spatial regions, have diverse magnitudes, and manifest various frequency characteristics. We refer to the collection of all visually observable motions in a scene as the *macro motion*. Factorizing and analyzing macro motions is essential in understanding and interpreting the dynamic world. We argue that analyzing macro motions requires a dynamic scene representation that bear three key properties: It should be able to represent macro motions faithfully; it should enable decomposing macro motions in both spatial domain and frequency domain; and it should allow representing 3D scenes.

While modeling motions have been a constant topic of interest in computer vision, graphics, and machine learning, existing methods do not meet the three properties simultaneously. Traditional motion processing and magnification methods allow analyzing motions by filtering input videos in frequency domain and editing the frequency components' magnitudes (Wu et al., 2012) or phases (Wadhwa et al., 2013). However, they only focus on micro motions (i.e., local and tiny motions) and they do not handle 3D scenes. Recent methods (Li et al., 2021; 2022a; Fridovich-Keil et al., 2023) exploit neural radiance fields (Mildenhall et al., 2021) to represent dynamic 3D scenes using deformation fields (Pumarola et al., 2021) or flow fields (Du et al., 2021). For example, D-NeRF (Pumarola et al., 2021) defines a template neural radiance field in the canonical space and builds correspondences between the observation space and the canonical space using a displacement field. Although 3D correspondences explicitly represent scene motions, they cannot be directly used for the motion analysis due to the lack of modeling underlying motion components.

In this work, we propose the Phase-based neural polynomial Gabor fields (Phase-PGF) that simultaneously meets all the three key properties for macro motion analysis. In particular, Phase-PGF represents a dynamic scene as a composition of wavelet-based neural fields, and the wavelet basis are modulated by a set of temporally-varying phases. Therefore, macro motions in dynamic scenes are captured by the phases. We show that under appropriate assumptions, Phase-PGF has theoretical properties that allow various macro motion analysis tasks, such as motion separation, motion

---

*Equal contribution.

smoothing, and motion intensity adjustment. We then instantiate Phase-PGF with a novel neural architecture and a training scheme for higher-quality dynamic scene representation and editing.

In our experiments, we collect examples of both 2D videos and 3D dynamic scenes (represented by multi-view videos). We show that Phase-PGF allows macro motion analysis on both 2D and 3D dynamic scenes, allowing several macro motion analysis applications including motion loop detection, motion separation, motion smoothing, and motion magnification.

In summary, our contributions are threefold: Firstly, we propose and formulate the problem of macro motion analysis. Secondly, we introduce Phase-based neural polynomial Gabor fields (Phase-PGF) for macro motion analysis. Lastly, our experiments show that Phase-PGF allows a series of dynamic scene editing tasks on both 2D and 3D scenes.

## 2 RELATED WORK

**Dynamic neural representations.** Using neural representations (Mildenhall et al., 2021; Sitzmann et al., 2020) to represent dynamic signals has emerged as a popular research topic in recent years. It has been widely used in representing 4D (3D space and 1D time) dynamic volumetric videos with advantages including high-fidelity rendering and low storage requirements. Some methods build explicit correspondences between frames by extracting dense flow or deformation (Pumarola et al., 2021; Du et al., 2021; Li et al., 2021; 2023b; Park et al., 2021a;b). However, such representation for motion is dense and is hard to analyze. Other approaches use inductive bias on scenes to assist the reconstruction of dynamic scenes(Peng et al., 2021; 2023; Weng et al., 2022), yet cannot be applied to general dynamic scenes. Recently some approaches propose to use hybrid representations (Fridovich-Keil et al., 2023; Shao et al., 2023) to implicitly represent dynamic 3D scenes, lacking interpretability on motion information. Beyond 3D dynamic scenes, there are also works focusing on using neural representations for 2D videos(Li et al., 2022b; Chen et al., 2021; Zhang et al., 2021b; Rho et al., 2022). However, most of them use implicit motion representation which cannot be easily analyzed or edited.

**Motion analysis and editing.** Motion analysis and editing in dynamic content is an important task in computer vision. Phase information is widely used in the field of motion analysis (Gautama & Van Hulle, 2002; Meyer et al., 2018; Mai & Liu, 2022). For motion editing, early work solves this problem with a Lagrangian perspective(Liu et al., 2005), yet renders artifacts with large motions. Wu et al. (2012) proposes to understand this task with an Eulerian perspective and Wadhwa et al. (2013) further proposes a phase-based method to perform motion magnification. Some other works (Davis et al., 2015b;a; Davis, 2016) use physical modal analysis to manipulate local motions. Despite being successful for tiny motions, they cannot make good editing on large and macro motions studied in this paper.

**Interpretable and editable neural scene representations.** With the development of neural representations (Mildenhall et al., 2021; Sitzmann et al., 2020), the interpretability of such methods has become an important research area. Several works on this topic focus on the frequency domain or the coarse-to-fine decomposition of the input signal (Lindell et al., 2022; Yang et al., 2022; Martel et al., 2021). Other works target at spatial decomposition of the input content (Shuai et al., 2022; Zhang et al., 2021a; Lu et al., 2020), allowing modifying the scene contents at an instance level. These prior works primarily solve static scenes, yet our work focuses on dynamic content. Moreover, our work focuses on the interpretability from the macro motion perspective of the representation, differing from prior works.

**Concurrent works.** Recently there have been some concurrent works on tiny motion analysis and editing. Feng et al. (2023) proposes a method to magnify local and tiny 3D motion, yet their method has challenges in manipulating *macro motion* discussed in this work. Li et al. (2023a) gives an approach to manipulate local motion, but their focus is also tiny motions.

## 3 APPROACH

In this section, we first formulate the problem of macro motion analysis. Then, we show an abstract formulation and theoretical properties of the proposed Phase-based neural polynomial Gabor

fields (Phase-PGF). Next, we introduce how we instantiate the formulation with neural networks to represent 2-D and 3-D dynamic signals. Finally, we present a training scheme for our approach.

## 3.1 MACRO MOTION ANALYSIS

We define the macro motion as follows:

**Definition 3.1** (Macro Motion). *We assume a dynamic scene can be decomposed into $k$ components $\{s_1, s_2, \cdots, s_k\}$, each with rigid motion $\mathbf{y}_i(t)$. The dynamic scene is represented using the following implicit function:*

$$s(\mathbf{x}, t) = \sum_i^k s_i(\mathbf{x} + \mathbf{y}_i(t)) = s_1(\mathbf{x} + \mathbf{y}_1(t)) + s_2(\mathbf{x} + \mathbf{y}_2(t)) + \cdots + s_k(\mathbf{x} + \mathbf{y}_k(t)), \quad (1)$$

*where we assume $s_i(\mathbf{x})$ is differentiable and non-constant over its domain. The macro motion is defined as $\mathcal{Y} = \{\mathbf{y}_1(t), \mathbf{y}_2(t), \cdots, \mathbf{y}_k(t)\}$.*

Note that this definition does not imply that a *cognitive object* is only represented using only *one* component $s_i$. Rather, an object can be represented using several components with several rigid motions, making it possible to model complex scenes without loss of generality.

We are interested in analyzing macro motion. Specifically, we aim to find a low-dimensional representation that represents $\mathcal{Y}$ and can be factorized in both the spatial domain and frequency domain.

## 3.2 PHASE-PGF: PHASE-BASED NEURAL POLYNOMIAL GABOR FIELDS

We propose Phase-based neural polynomial Gabor fields (Phase-PGF). In particular, to inherently allow factorization of the represented signals in both spatial and frequency domains, we adopt polynomial neural fields (Yang et al., 2022) with Gabor basis functions (Feichtinger & Strohmer, 2012) as the basic building block. To represent motions in a low-dimensional space, we leverage the phases of the Gabor basis functions. The formal definition is given by:

**Definition 3.2** (Phase-PGF). *Let $\mathcal{B} = \{g_1, g_2, \cdots, g_m\}$ to be a shift-orthogonal[1] set of Gabor functions $g_i(\mathbf{x}; \gamma_i, \mu_i, \omega_i, \phi_i) = \exp(-\frac{\gamma_i}{2}\|\mathbf{x} - \mu_i\|_2)\sin(\omega_i \mathbf{x} + \phi_i)$. Let $\mathcal{H} = \{h_1(t), h_2(t), \cdots, h_m(t)\}$ be a set of phase functions where $h_i(t) : \mathbb{R}^1 \to \mathbb{R}^1$. A Phase-based neural polynomial Gabor fields (Phase-PGF) is a neural network $f(\mathbf{x}, t) = p_L \circ p_{L-1} \circ \cdots \circ p_1 \circ \Psi(\mathbf{x}, t)$, where $\forall i$, $p_i$ are finite degree multivariate polynomials, and $\Psi(\mathbf{x}, t) = \{g_1(\mathbf{x} + h_1(t)), g_2(\mathbf{x} + h_2(t)), \cdots, g_m(\mathbf{x} + h_m(t))\}$ is $m$-dimensional feature encoding using basis $\mathcal{B}$ and phase functions $\mathcal{H}$.*

Intuitively, the time-varying phase information allows our model to represent the macro motions using a low-dimensional vector. Therefore, the macro motions can be analyzed and manipulated by controlling the phase information.

We then theoretically discover the properties of the proposed representation. First, we show that the phases can be used to analyze the periodic components of the macro motion in the scene.

**Theorem 3.1** (Periodicity Correlation). *For a Phase-PGF with phase functions $\mathcal{H}$ already trained to represent a dynamic scene $s(\mathbf{x}, t)$, if there exists $T > 0$ such that some motion $\mathbf{y}_i(t + T) = \mathbf{y}_i(t)$, then $\exists h_j \in \mathcal{H}, h_j(t + T) = h_j(t)$.*

We include the proofs of all theorems in appendix A. This theorem indicates that periodic motion components can correspond to components of the phase information.

If a scene has complicated composed motions, e.g., a scene with multiple dynamic objects, Phase-PGF allows unsupervisedly factorizing different motion components into different components in the phases. Specifically, we have the following property:

**Theorem 3.2** (Motion Separation). *Assume that a scene has several objects with motion $\mathbf{y}_i(t)$ associated with each object $i$. For each component, the implicit functions representing object $i$ lies in*

---

[1]$\langle g_i(\mathbf{x} + \alpha), g_j(\mathbf{x} + \beta) \rangle = \delta_{ij}$, where $\delta$ is the Kronecker delta function.

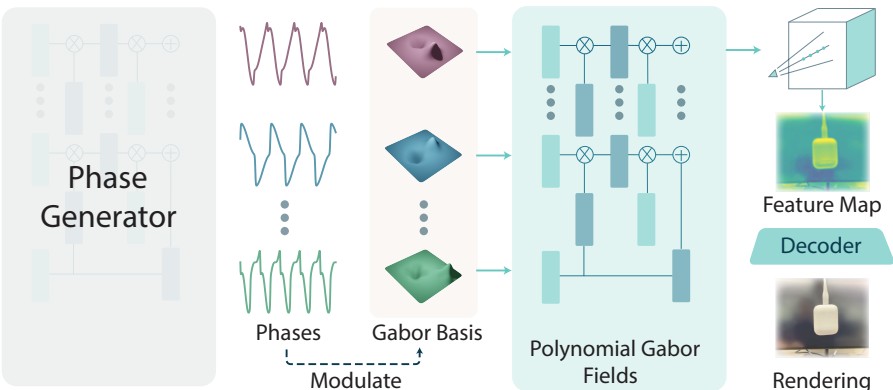

Figure 1: **Overview of Phase-PGF.** A phase generator yields time-varying phases to modulate the Gabor basis in Phase-PGF, which produces a latent feature field for volume rendering. Please refer to the text in Sec. 3.3 for more descriptions.

*the span of $\mathcal{B}_i$. Then, a Phase-PGF $f(\mathbf{x}, t)$ can be decomposed into several Phase-PGFs:*

$$f(\mathbf{x}, t) = \sum_i^N f_i(\mathbf{x}, t), \tag{2}$$

*where $f_i(\mathbf{x}, t)$ is a Phase-PGF with basis $\mathcal{B}_i$.*

This theorem indicates that it is possible to disentangle the scene macro motion based on the phases. For example, if the macro motion can be separated into high-frequency components and low-frequency components, or if the macro motion contains motion components present at different spatial regions, then by filtering out the desired phase components, we can manipulate the scene motion in a factorized manner.

We have now shown that the extracted phase information from the Phase-PGF can be used to observe the properties of the macro motion. We then show that this representation can be edited to manipulate the scene motion.

**Theorem 3.3** (Motion Smoothing). *Assume that the a dynamic scene is represented by a trained Phase-PGF with phases $\mathcal{H}$. If we apply a low-pass filter on some $h_j$ that corresponds to some motion $\mathbf{y}_i$, then the motion would be smoother, i.e., $|\frac{d}{dt}\mathbf{y}_i(t)|$ is attenuated.*

Intuitively, motion smoothing corresponds to attenuating high-frequency motion components. Besides motion filtering, it is also possible to perform the manipulation in the dimension of motion intensity/magnitude:

**Theorem 3.4** (Motion Intensity Adjustment). *Assume that a scene component motion takes the form $\mathbf{y}_i(t) = e^{it}$ and the scene is represented using a Phase-PGF with a phase $h_j(t)$ corresponds to $\mathbf{y}_i(t)$. If we multiply the phase by a coefficient $A$, then the scene motion would be manipulated to $\mathbf{y}_i(t) = Ae^{it}$.*

We include the proofs of all the theorems in appendix A.

### 3.3 INSTANTIATION OF PHASE-PGF

In the previous section, we have introduced the abstract definition of Phase-PGF and its properties. In this section, we further illustrate how we empirically instantiate Phase-PGF to represent 2D and 3D dynamic scenes. We show an overview in Figure 1, where we have a phase generator (on the left of Figure 5) that generates the $m$ time-varying phases $\mathcal{H} = \{h_1(t), h_2(t), \cdots, h_m(t)\}$ for modulating the Gabor basis $\mathcal{B} = \{g_1, g_2, \cdots, g_m\}$. The Phase-PGF $f(\mathbf{x}, t) = p_L \circ p_{L-1} \circ \cdots \circ p_1 \circ \Psi(\mathbf{x}, t)$ generates a latent feature map and a neural latent decoder (detailed below) renders an image from the feature map.

**Phase Generator for** $\mathcal{H}$**.** Since the phases are central to our model, the phase generator architecture should ideally be expressive and structured. To this end, we instantiate each $h_i$ with a Polynomial Neural Field (PNF) (Yang et al., 2022), which allows decomposing the phases into frequency sub-bands. The frequency decomposition of phases allows easy manipulation over the phases, thus over the macro motions.

**Neural Latent Decoder.** While Phase-PGF takes the form of $f(\mathbf{x}, t) = p_L \circ p_{L-1} \circ \cdots \circ p_1 \circ \Psi(\mathbf{x}, t)$ where $p$ denotes a polynomial, we empirically find that it consumes a large amount of GPU memory if we directly use $f(\mathbf{x}, t)$ to represent raw pixels in complex and high-resolution scenes. Therefore, we propose to couple Phase-PGF with a neural latent decoder. That is, Phase-PGF represents a low-resolution latent feature field of the dynamic scenes, and the neural latent decoder decodes the latent features into high-resolution results, similar to the feature fields in prior works (Niemeyer & Geiger, 2021).

More specifically, for representing 2D videos, we first use Phase-PGF $f(\mathbf{x}, t)$ to render a 2D feature map $m$. Then, we use a 2D CNN decoder to render the final image $\hat{I} = D(m)$. For 3D dynamic scenes, to render a frame from a viewpoint, we adopt the volume rendering (Mildenhall et al., 2021) formation. That is, we do ray marching to get a set of sampled points on the rays of the frame, and then we use volume rendering to aggregate the sampled features to form the feature map. We also use a 2D CNN to decode the feature map into a final RGB image.

The neural latent decoder $D$ is designed in a pyramid manner. We first decode the feature into a low-resolution RGB image $I_L = D_L(m)$. Then, we use another decoder $D_H$ to obtain the final high-resolution output $\hat{I} = D_H(I_L)$.

We refer the readers to appendix C.1 and appendix C.2 for more details on the neural network implementation and the phase space.

## 3.4 TRAINING PHASE-PGF TO REPRESENT DYNAMIC SCENES

**Multi-stage Training.** We train Phase-PGF in a multi-stage manner. In the first stage, we supervise both the learning of Phase-PGF $F$ and decoder $D_L$ with reconstruction loss $\mathcal{L}_1 = ||\hat{I}_L - I_L||_2$, where $I_L$ is the ground-truth low-resolution image.

In the second stage, we further improve the high-resolution detail of the rendered image with patch-based sampling and perceptual loss (Johnson et al., 2016), during each training step, we randomly sample a patch from the high-resolution image, and the loss $\mathcal{L}_2$ is defined as $\mathcal{L}_2 = ||\hat{I} - I||_2 + ||m_{\mathrm{vgg}}(\hat{I}) - m_{\mathrm{vgg}}(I)||_2$, where $I$ is the patch from the ground-truth high-resolution image and $m_{\mathrm{vgg}}$ is a feature map extracted by a pre-trained VGG network (Johnson et al., 2016).

**Adversarial Training.** The multi-stage training enables high-fidelity reconstruction of the input scenes. However, extrapolating macro motions (e.g., for motion magnification) requires further regularization on the neural latent decoder. Therefore, we propose to further use adversarial training after the two stages. In particular, given the generated phases $\mathcal{H}$, we perform simple magnification to it and apply these augmented phases to the decoders. We want the decoder to generate plausible images even when the input phases are extrapolated. Specifically, for a given phase sequence $h(t) \in \mathcal{H}$, the augmented $h'(t|\lambda, b_l, b_h)$ is defined as follows:

$$h'(t|\lambda, b_l, b_h) = h(t) + (\lambda - 1) \cdot y(t|b_l, b_h), \tag{3}$$

$$y(t|b_l, b_h) = \mathcal{F}^{-1}(T(\mathcal{F}(h(t)), b_l, b_h)), \tag{4}$$

where $\lambda$ is intensity manipulation coefficient, $b_l$ and $b_h$ are subband limits for specific component of the signal, $\mathcal{F}$ and $\mathcal{F}^{-1}$ are fourier transform and its inversion, and $T(f)$ is a band-limit filter defined as follows:

$$T(f) = \begin{cases} 1, & \text{if } f_1 \leq f \leq f_2, \\ 0, & \text{otherwise.} \end{cases} \tag{5}$$

We additionally construct a discriminator $\mathcal{D}$ to judge whether the image decoded from the extrapolated phase sequence is similar to the input sequence. The adversarial loss is defined as follows:

$$\mathcal{L}_{\mathrm{adv}} = \mathbb{E}_{\mathbf{x}, t}[\log \mathcal{D}(f(\mathbf{x}, t)] + \mathbb{E}_{\mathbf{x}, t, \lambda}[\log(1 - \mathcal{D}(\mathcal{T}_{f, \lambda}(\mathbf{x}, t)], \tag{6}$$

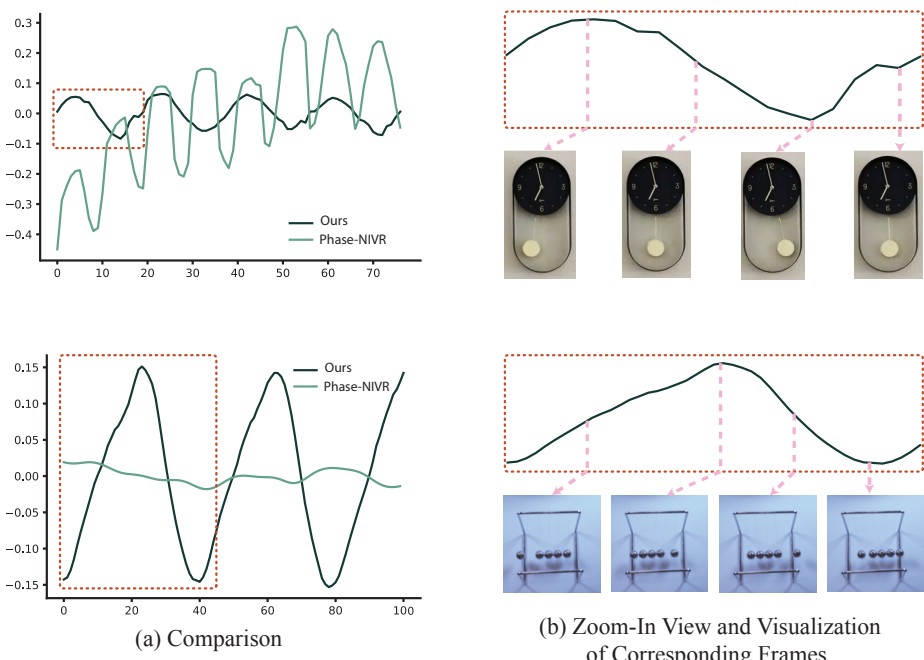

(a) Comparison

(b) Zoom-In View and Visualization
of Corresponding Frames

Figure 2: **On the interpretability of motion representations of Phase-PGF and Phase-NIVR.**
**(a)** Given the 2D videos in (b), we compare the representative phases generated by our method and
Phase-NIVR. **(b)** We zoom in on the phase generated by our model and visualize some keyframes
to illustrate the correspondence between the macro motion and the generated phase.

| Sequence Name | Prefer Ours |
|---|---|
| Clock | 76.5% |
| Pendulum | 81.4% |
| Balls | 90.2% |
| Kid | 71.6% |
| Bouncing | 76.5% |
| **Average** | 79.2% |

|  | Ours | Phase-NIVR (Mai & Liu, 2022) |
|---|---|---|
| **Damping** | 0.999 | 0.015 |
| **Projectile (X)** | 0.998 | 0.793 |
| **Projectile (Y)** | 0.859 | 0.066 |

Table 1: **Human preference study on the interpretability of generated phases.** The phases from ours are more interpretable than the baseline.

Table 2: **Quantitative comparison using normalized cross-correlation score on phases in synthetic datasets.** Projectile(X) and Projectile(Y) mean motion in two different spatial dimensions.

where $\mathcal{T}_{f,\lambda}$ denotes a functional that augment the phase generated by $f$ with a motion intensity adjustment coefficient $\lambda$, as defined above. For more details on training, please refer to appendix C.4.

## 4 EXPERIMENTS

To evaluate our approach on the task of macro motion analysis, we collect several examples including both 2D and 3D dynamic scenes. We show that our Phase-PGF learns interpretable motion representations to allow macro motion analysis, which is demonstrated by motion loop detection and motion separation. Afterward we show that Phase-PGF allows macro motion editing tasks including motion intensity adjustment and motion smoothing.

### 4.1 EXAMPLES

We collect several dynamic scene examples including both 2D and 3D scenes.

**Synthetic 2D videos.** We render several different videos with varying motion complexities with ground truth motion to quantitatively evaluate the proposed method. Please refer to appendix D.1.

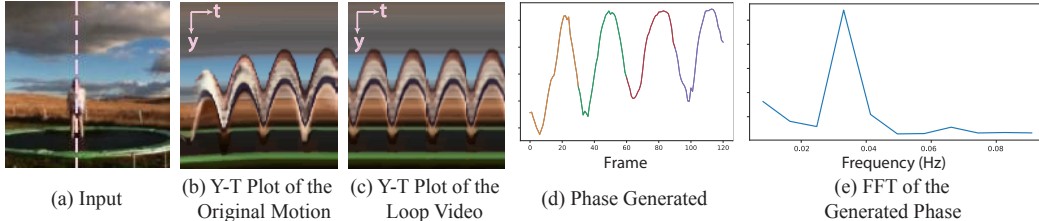

| (a) Input | (b) Y-T Plot of the Original Motion | (c) Y-T Plot of the Loop Video | (d) Phase Generated | (e) FFT of the Generated Phase |

Figure 3: **Loop detection using extracted phase information.** Given the (a) input video, we first learn the (d) phase to represent the macro motion. We then perform (e) FFT on the phase to get the frequency domain on the phase and get the periodicity information. Then the phase was segmented based on the period which is shown as different colors in (d). The seamless video (c) can be then made according to this information and we show the corresponding Y-T plots in (b) and (c).

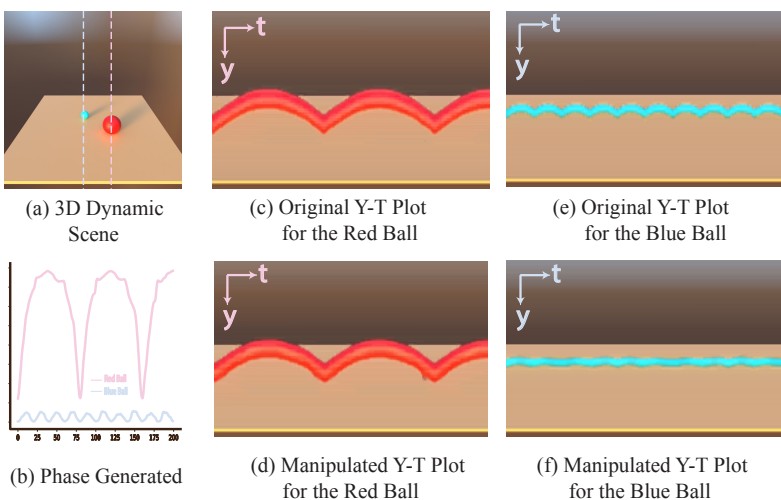

| (a) 3D Dynamic Scene | (c) Original Y-T Plot for the Red Ball | (e) Original Y-T Plot for the Blue Ball |
| (b) Phase Generated | (d) Manipulated Y-T Plot for the Red Ball | (f) Manipulated Y-T Plot for the Blue Ball |

Figure 4: **Motion separation.** This 3D scene contains two balls with different motions. Our method is able to decompose their motion using different phases, as shown in (b). We attempt to show that we are able to keep the red ball's motion **unchanged** while modifying the motion of the blue ball. (c) and (d) show that the motion of the red ball remains unchanged after modifying. (e) and (f) show that we successfully attenuate the motion of the blue ball.

**Real 2D videos.** We also collect several real videos from the Internet and from our own capture to show the generalizability and applicability of the proposed Phase-PGF. We take some videos from Mai & Liu (2022) and other Internet sources to form a dataset of real 2D videos.

**3D Dynamic Scene.** Our Phase-PGF can also be used to model 3D dynamic scenes with neural rendering. To demonstrate this, we render a synthetic 3D dynamic scenes with two different balls bouncing on a table top.

## 4.2 MACRO MOTION ANALYSIS: LOOP DETECTION AND MOTION SEPARATION

For macro motion analysis, a preliminary requirement is that the motion representation is interpretable, i.e., the components of the factorized motion representation should be able to establish correspondences to scene motion components. To do this, we compare the phase generated by our Phase-PGF against a previous method Phase-NIVR (Mai & Liu, 2022). Phase-NIVR also aims to generate phases for input motion, yet it does not allow representing 3D scenes. We show a comparison of phases by both methods in Figure 2 (a), and show keyframes corresponding to our generated phases in Figure 2 (b).

From Figure 2, we observe that our Phase-PGF generates a phase sequence that aligns with the motion more coherently. A possible reason is that Phase-NIVR uses the SIREN network (Sitzmann et al., 2020) as the scene representation which does not support decomposing different components

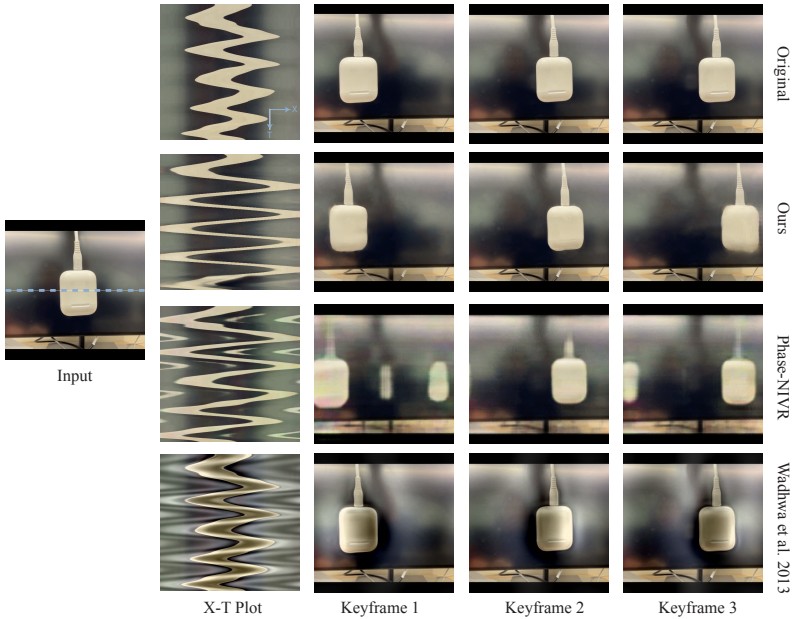

Figure 5: **Comparison of the result of motion intensity adjustment between different baselines and the proposed method.** In the first column, we show the different X-T plots generated by concatenating the strips indicated by the blue line in the input video. In the following three columns, we visualize the rendered results in different keyframes of the input motion.

of the dynamic scene. Please check out the supplementary website at `https://chen-geng.com/phasepgf` for video results.

To quantitatively evaluate the motion representuation interpretability, we conduct a human preference study. We show several videos and the generated phases by both methods, and ask the participants to pick one that they believe better explains the motions in the videos. We use a research-oriented cloud source platform[2]. We show the human study results in Table 1. From Table 1, we see that the phases generated by Phase-PGF are believed to better explain the motions.

We additionally evaluate the **normalized cross-correlation** of phases generated by our method and the baseline on datasets with ground-truth motion trajectory. The results are shown in Table 2. Please refer to appendix D.2 for more details on this.

To discover more complex motion patterns, we synthesize and collect several videos with varying motion complexities and evaluate them in appendix D.1.

We further show two macro motion analysis applications: motion loop detection and motion separation.

**Motion loop detection.** In Figure 3, we show an example of motion loop detection on the input video. To do this, we identify the periodicity of the input video, and then we perform phase fusing (Mai & Liu, 2022) to generate an infinitely looping video.

**Motion separation.** In Figure 4, we show motion separation on a synthetic 3D scene, where two balls bounce at different speeds. Phase-PGF is able to decompose the macro motion using a frequency filter. Specifically, we use band-limit filters on the generated phase sequence, and then we isolate the motion of interest (the motion of the blue ball) according to the frequency. In this way, we can manipulate (mollify) the motion of the blue ball without affecting the motion of the red ball.

### 4.3 MACRO MOTION EDITING: INTENSITY ADJUSTMENT AND SMOOTHING

Besides analysis, our Phase-PGF also allows editing macro motions. Specifically, we consider two motion editing tasks: motion intensity adjustment and motion smoothing. Please refer to appendix C.3 for details on the editing operations.

---

[2]This human preference study form can be found at the appendix.

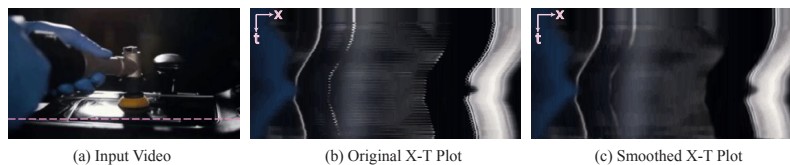

| (a) Input Video | (b) Original X-T Plot | (c) Smoothed X-T Plot |

Figure 6: **Motion smoothing.** We smooth the motion in the video by removing high-frequency components in the phase. (b) and (c) shows the X-T plot of a slice in the original and edited video.

|  | Ours | Phase-NIVR | Wadhwa et al. (2013) |
|---|---|---|---|
| Motion Magnification Score | **3.30** | 3.06 | 2.24 |
| Visual Quality Score | **3.14** | 1.76 | 1.88 |
| FID | **0.512** | 10.255 | 5.889 |

Table 3: **Quantitative comparison on the result of motion intensive adjustment.** For the user study part (First two rows) We ask the user to give two scores for each video generated by the corresponding method. The first score is asked to evaluate how large the visual motion magnification intensity is (1 for identical and 5 for the largest). The second score is asked to evaluate the visual quality of the rendering (1 for the worst and 5 for perfect). We additionally evaluate the FID score for the videos generated by different methods, as shown in the third row.

**Motion Intensity Adjustment.** Phase-PGF allows modifying the intensity or magnitude of the macro motion in the represented dynamic scenes. Following the method introduced in Sec. 3.4, we show motion magnification results in Figure 5. We compare our method with a traditional motion magnification method Wadhwa et al. (2013) and Phase-NIVR. From Figure 5 we observe that Phase-PGF allows magnification with much fewer artifacts. In comparison, Phase-NIVR generates a temporally flickering video with ghosting artifacts, and Wadhwa et al. (2013) failed to magnify the macro motion as it focuses on tiny motions. Our observation is further consolidated by a human preference study, where we show the results in Table 3.

We also evaluate the Fréchet Inception Score for manipulated videos generated from different methods. The result is shown in Table 3. It can be seen that our method surpasses other baselines greatly in terms of visual quality. Please refer to appendix D.2 for more details.

We perform ablation studies to prove the effectiveness of the introduced components. Please refer to appendix D.3 for detailed discussion.

**Motion Smoothing.** We show that Phase-PGF also allows motion smoothing. In Figure 6, we show that by removing high-frequency bands of the generated phase sequence, we can perform motion smoothing on the macro motion. From Figure 6, we observe that the high-frequency flickering of the paddle has been removed, leading to a smoother motion.

## 5 CONCLUSION

In this work, we propose and formulate the problem of macro motion analysis. We propose Phase-based neural polynomial Gabor fields (Phase-PGF) that represents motions in dynamic scenes with generated phase sequences. We show that Phase-PGF allows multiple macro motion analysis and editing tasks, including loop detection, motion separation, motion magnification, and motion smoothing.

**Limitations.** One limitation is that Phase-PGF shows slight artifacts in boldly magnifying large motions. Another limitation is that Phase-PGF currently does not scale well to complex large-scale 3D dynamic scenes due to computational efficiency (we need more Gabor basis in larger scenes). This might be addressed by spatially adaptive Gabor basis. We also discuss other failure cases at appendix D.7.

## ACKNOWLEDGMENTS

This work is in part supported by the Stanford Institute for Human-Centered AI (HAI) and Samsung.

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

## A  DEFINITIONS AND PROOFS

In this section, we provide proofs of the theorems in the main paper. Our Phase-PGF is based on the polynomial neural fields (Yang et al., 2022). In the following proofs, we will use some of the properties of the polynomial neural fields. We refer the reader to Yang et al. (2022) for more context.

*Proof of theorem 3.1.* Let's assume that the scene at $t = 0$ is represented with the implicit function $s(\mathbf{x})$, and the active content is doing rigid motion $\mathbf{y}(t)$. Then the dynamic scene to be represented can be denoted as $s(\mathbf{x} + \mathbf{y}(t))$, where both $s(\mathbf{x})$ and $\mathbf{y}(t)$ are unknown.

Now the dynamic content is fit using a Phase-PGF $f(\mathbf{x}, t)$, which gives:

$$f(\mathbf{x}, t) = s(\mathbf{x} + \mathbf{y}(t)), \quad \forall \mathbf{x}, t. \tag{7}$$

If we have $\mathbf{y}(t + T) = \mathbf{y}(t)$ holds for any $t$, then:

$$f(\mathbf{x}, t + T) = s(\mathbf{x} + \mathbf{y}(t + T)) \tag{8}$$
$$= s(\mathbf{x} + \mathbf{y}(t)) \tag{9}$$
$$= f(x, t). \tag{10}$$

Consider the property of Polynomial functions(Yang et al., 2022), we have:

$$f(\mathbf{x}, t) = \sum_i^m \omega_i g_i(\mathbf{x} + h_i(t)), \tag{11}$$

then we can have:

$$\sum_i^m \omega_i(g_i(\mathbf{x} + h_i(t + T)) - g_i(\mathbf{x} + h_i(t))) = 0, \quad \forall \mathbf{x}, t. \tag{12}$$

Consider doing inner-product with $g_j$, we have:

$$\sum_i^m \omega_i \langle (g_i(\mathbf{x} + h_i(t + T)) - g_i(\mathbf{x} + h_i(t))), g_j(\mathbf{x}) \rangle \tag{13}$$

$$= \sum_i^m \omega_i \langle g_i(\mathbf{x} + h_i(t + T)), g_j(\mathbf{x}) \rangle - \langle g_i(\mathbf{x} + h_i(t)), g_j(\mathbf{x}) \rangle \tag{14}$$

$$= \omega_j \langle g_j(\mathbf{x} + h_j(t + T)) - g_j(\mathbf{x} + h_j(t)), g_j(\mathbf{x}) \rangle \tag{15}$$
$$= 0 \tag{16}$$

Therefore, we have $g_j(\mathbf{x} + h_j(t + T)) = g_j(\mathbf{x} + h_j(t))$, which implies $h_j(t + T) = h_j(t)$.

$\square$

*Proof of theorem 3.2.* According to the assumption, the dynamic scene can be represented using the following implicit function:

$$g(\mathbf{x}, t) = \sum_i s_i(\mathbf{x} + \mathbf{y}_i(t)), \tag{17}$$

where $s_i$ is in the span of $\mathcal{B}_i$. Also, we have:

$$\sum_i^k s_i(\mathbf{x} + \mathbf{y}_i(t)) = f(\mathbf{x}, t). \tag{18}$$

Considering the property of polynomial fields(Yang et al., 2022), the $f(\mathbf{x}, t)$ can be written as:

$$f(\mathbf{x}, t) = \sum_i^m \omega_i g_i(\mathbf{x} + h_i(t)), \tag{19}$$

Then we have:

$$\sum_i^k s_i(\mathbf{x} + \mathbf{y}_i(t)) = \sum_i^m \omega_i g_i(\mathbf{x} + h_i(t)) \tag{20}$$

$$= \sum_i^k \sum_{g_j \in \mathcal{B}_i} \omega_j g_j(\mathbf{x} + h_i(t)) \tag{21}$$

Thus we can get:

$$s_i(\mathbf{x} + \mathbf{y}_i(t)) = \sum_{g_j \in \mathcal{B}_i} \omega_j g_j(\mathbf{x} + h_i(t)), \quad \forall i \tag{22}$$

Let's denote $f_i(\mathbf{x}, t) = \sum_{g_j \in \mathcal{B}_i} \omega_j g_j(\mathbf{x}, t)$, then we have got a decomposition of $f(\mathbf{x}, t) = \sum f_i(\mathbf{x}, t)$, where each $f_i$ is supported by basis $\mathcal{B}_i$. $\qquad \square$

*Proof of theorem 3.2.* According to the assumptions, if the rigid motion is represented using $\mathbf{y}(t)$, then:

$$s(\mathbf{x} + \mathbf{y}(t)) = f(\mathbf{x}, t), \tag{23}$$

$$= \sum \omega_i g_i(\mathbf{x} + h_i(t)), \quad \forall \mathbf{x}, t. \tag{24}$$

We have used the property of the polynomial fields(Yang et al., 2022) in the derivation above. We first perform Fourier decomposition of implicit function $s(\mathbf{x})$ and the basis $g_i(x)$, so that we can get:

$$s(x) = \int_\infty G(f) e^{i 2\pi f x} \mathrm{d}f \tag{25}$$

$$g_i(x) = \int_\infty g_i'(f) e^{i 2\pi f x} \mathrm{d}f \tag{26}$$

By modulating the equation above, we can obtain:

$$s(\mathbf{x} + \mathbf{y}(t)) = \int_\infty G(f) e^{i 2\pi f (\mathbf{x} + \mathbf{y}(t))} \mathrm{d}f \tag{27}$$

$$= \sum \omega_i g_i(\mathbf{x} + h_i(t)), \tag{28}$$

$$= \sum \omega_i \int_\infty \hat{g}_i(f) e^{i 2\pi f (\mathbf{x} + h_i(t))} \mathrm{d}f, \quad \forall \mathbf{x}, t \tag{29}$$

We perform the Fourier transform of the equation above.

$$G(f)e^{i2\pi fy(t)} = \sum \omega_i \hat{g}_i(f)e^{i2\pi fh_i(t)}, \quad \forall f. \tag{30}$$

Then we have:

$$y(t) = \frac{1}{2\pi i f} \ln[\frac{1}{G(f)} \sum \omega_i \hat{g}_i(f)e^{i2\pi fh_i(t)}]. \tag{31}$$

By performing low-pass filter on $h_i(t)$, we can obtain a new phase $h_L(t)$ with attenuated high-frequency information, which has the property:

$$|\frac{\mathrm{d}}{\mathrm{d}t}h_L(t)| \le |\frac{\mathrm{d}}{\mathrm{d}t}h_i(t)| \tag{32}$$

We consider the derivative of the corresponding $y_L(t)$ to evaluate the change in its smoothness.

$$|\frac{\mathrm{d}}{\mathrm{d}t}y_L(t)| = |\frac{\mathrm{d}}{\mathrm{d}t}\frac{1}{2\pi i f} \ln[\frac{1}{G(f)} \sum \omega_i \hat{g}_i(f)e^{i2\pi fh_L(t)}]| \tag{33}$$

$$= |\frac{1}{2\pi i f}\frac{\mathrm{d}}{\mathrm{d}t} \ln[\frac{1}{G(f)} \sum \omega_i \hat{g}_i(f)e^{i2\pi fh_L(t)}]| \tag{34}$$

$$= |\frac{G(f)}{2\pi i f \cdot \sum \omega_i \hat{g}_i(f)e^{i2\pi fh_L(t)}}\frac{\mathrm{d}}{\mathrm{d}t}[\frac{1}{G(f)} \sum \omega_i \hat{g}_i(f)e^{i2\pi fh_L(t)}]| \tag{35}$$

$$= |\frac{1}{2\pi i f \cdot \sum \omega_i \hat{g}_i(f)e^{i2\pi fh_L(t)}}\frac{\mathrm{d}}{\mathrm{d}t} \sum \omega_i \hat{g}_i(f)e^{i2\pi fh_L(t)}| \tag{36}$$

$$= |\frac{1}{2\pi i f \cdot \sum \omega_i \hat{g}_i(f)e^{i2\pi fh_L(t)}} \sum \omega_i \hat{g}_i(f)\frac{\mathrm{d}}{\mathrm{d}t}e^{i2\pi fh_L(t)}| \tag{37}$$

$$= |\frac{1}{2\pi i f \cdot \sum \omega_i \hat{g}_i(f)e^{i2\pi fh_L(t)}} \sum \omega_i \hat{g}_i(f)e^{i2\pi fh_L(t)}i2\pi f\frac{\mathrm{d}}{\mathrm{d}t}h_L(t)| \tag{38}$$

$$= \frac{|\sum \omega_i \hat{g}_i(f)\frac{\mathrm{d}}{\mathrm{d}t}h_L(t)|}{|\sum \omega_i \hat{g}_i(f)|} \tag{39}$$

$$\le \frac{|\sum \omega_i \hat{g}_i(f)\frac{\mathrm{d}}{\mathrm{d}t}h_i(t)|}{|\sum \omega_i \hat{g}_i(f)|} \tag{40}$$

$$= |\frac{\mathrm{d}}{\mathrm{d}t}y(t)|. \tag{41}$$

Therefore, we can have that the $y_L(t)$ is smoother than $y(t)$, which finishes the proof.

$\square$

*Proof of theorem 3.4.* According to the assumption, we have:

$$s(\mathbf{x} + e^{it}) = f(\mathbf{x}, t) \tag{42}$$

$$= \sum \omega_i g_i(\mathbf{x} + h_i(t)) \tag{43}$$

Since we assume $s(\mathbf{x})$ is in the span of $\mathcal{B}$, it can decomposed into:

$$s(\mathbf{x}) = \sum \gamma_i g_i(\mathbf{x}) \tag{44}$$

Then:

$$\sum \gamma_i g_i(\mathbf{x} + e^{it}) = \sum \omega_i g_i(\mathbf{x} + h_i(t)) \tag{45}$$

According to the shift-orthogonality of $\mathcal{B}$,

$$h_i(t) = e^{it} \tag{46}$$

If we perform motion intensity adjustment to $h_i(t)$ with magnitude $A$, then we can get modified phase $h'(t) = Ah_i(t)$, so that the scene motion can be modified to:

$$h'(t) = Ah_i(t) \tag{47}$$
$$= Ae^{it}. \tag{48}$$

$\square$

## B  HUMAN PREFERENCE STUDY

We conduct user studies using the Prolific platform.

For the user study shown in Table 1, we have collected 102 effective responses. We designed our form using Google Forms. In the form, we asked the user to compare 5 pairs of videos with plots trying to explain the motion inside the video. For each pair, we asked the user the following prompt: *In which video (left or right), does the bottom plot better explain the motion in the top video?* The user will choose between two answers and we collect the data and make statistical analysis. The video pairs can be found at our supplementary website: `https://chen-geng.com/phasepgf`.

For the user study shown in Table 3, we have collected 110 effective responses. In the study, we first asked the users to watch three video pairs from three methods (each pair consists of a top video and a bottom video). In each pair, the bottom video is trying to magnify the motion in the top video. Then for each pair, we asked them two questions. The first question is: *The bottom video tries to magnify the motion of the white object in the top video. Do you think the bottom video successfully magnifies the motion? Please rate on a scale of 1 to 5, where 1 means "Not at all" and 5 means "Very much so".* The answer to this question was used to calculate the Motion Magnification score. The second question is: *You are viewing the same video pair as before. Please evaluate the visual quality of the bottom video. On a scale of 1 to 5, rate the quality of the bottom video.* The answer to this question was used to calculate the Visual Quality Score. The videos can be found at our supplementary website: `https://chen-geng.com/phasepgf`.

## C  METHOD DETAILS

In this section, we provide more details on the proposed method. The code will be released upon publication.

### C.1  NETWORK ARCHITECTURE

We first elaborate on more details on the network architecture. In the following, we discuss specific design choices related to different modules in the proposed framework.

**Phase Generator.**  The key aspects of an ideal phase generator are expressiveness and controllability. To achieve these two goals, we use a polynomial neural field (Yang et al., 2022) conditioned on $t$ to serve as the backbone. Specifically, we follow the architecture described in Yang et al. (2022), where the basis is defined using Fourier waves.

Each polynomial neural field takes normalized timestamp $t$ as input and outputs a one-dimension scalar $h(t)$ as the phase in the timestamp $t$. Following Yang et al. (2022), $h(t)$ is defined as below:

| Hyper-Parameter Name | Value |
|---|---|
| Output Dimension | 1 |
| Hidden Dimension | 8 |
| Number of Bandwidths | 4 |

Table 4: Hyper-parameters used in Phase Generator.

$$h(t) = \sum_j F_j(t), \tag{49}$$

where $F_j(t)$ is a sub-PNF for one subband. $F_j(t)$ is further implemented using the following factorization:

$$F_j(t) = \tanh(\sum_{k=1}^{n} G_j(t, b_k, b_k) W_{jk} Z_{j,k}(t)), \tag{50}$$

$$Z_{j,1}(t) = G_j(t, 0, \Delta_1), \tag{51}$$

$$Z_{j,k}(t) = G_j(t, 0, \Delta_k) W_i Z_{j,k-1}(t), \tag{52}$$

where $G_j(t, a, b)$ is a subband limited in $R^{(\infty)}(a, b, d(\theta_j), \delta)$, $\Delta_k = b_k - b_{k-1}$. $G_j(t, a, b)$ is expressed in the linear combination of basis sampled from the defined subband:

$$G_j(t, a, b) = \mathbf{W}_k \gamma_j(t), \gamma_j \in R^{(\infty)}(a, b, d(\theta_j), \delta)^d, \mathbf{W}_k \in \mathbb{R}^{h \times d}, \tag{53}$$

where $\gamma_j$ is Fourier basis.

Each polynomial neural field $h(t)$ represents a continuous one-dimensional phase. To represent the whole phase space of dimension $K$, we instantiate an individual polynomial neural field $h_i(t)$ for each phase in the phase space. Then we ensemble them to form the whole phase.

We list the hyper-parameters used to instantiate our phase generator in Table 4.

**Spatial Polynomial Gabor Fields.** Previously we have discussed the design of the phase generator, which is a 1-D function taking in time input and output a dynamic phase sequence. We then discuss the design of the polynomial Gabor field which represents the scene spatially.

The major difference between the architecture of temporal polynomial neural fields (T-PNF) and spatial polynomial Gabor fields(S-PGF) lies in the definition of $G_j$. Specifically, S-PGF is also realized as the summation of different S-PGFs representing different subbands:

$$F(\mathbf{x}) = \sum_j F_j(\mathbf{x}), \tag{54}$$

$$F_j(\mathbf{x}) = \sum_{k=1}^{n} \mathcal{G}_j(\mathbf{x}, b_k, b_k) W_{jk} Z_{j,k}(\mathbf{x}), \tag{55}$$

$$Z_{j,1}(\mathbf{x}) = \mathcal{G}_j(\mathbf{x}, 0, \Delta_1), \tag{56}$$

$$Z_{j,k}(\mathbf{x}) = \mathcal{G}_j(\mathbf{x}, 0, \Delta_k) W_i Z_{j,k-1}(\mathbf{x}), \tag{57}$$

where $\mathcal{G}_j(\mathbf{x}, a, b)$ is a subband limited in $R^{(\infty)}(a, b, d(\theta_j), d(\mu_j), d(\gamma_j), \delta)$. It is further defined using Gabor basis:

$$G_j(\mathbf{x}, a, b) = \mathbf{W}_k g_j(\mathbf{x}), g_j \in R^{(\infty)}(a, b, d(\theta_j), d(\mu_j), d(\gamma_j), \delta)^d, \mathbf{W}_k \in \mathbb{R}^{h \times d}, \tag{58}$$

where $g_j$ is Gabor basis. The hyper-parameters in this model are listed in Table 5.

| Hyper-Parameter Name | Value |
|---|---|
| Output Dimension | 32 (2D Video) / 33 (3D Scene) |
| Hidden Dimension | 64 |
| Number of Bandwidths | 4 |

Table 5: Hyper-parameter used in Spatial Polynomial Gabor Fields.

| Layer Type | In Ch | Out Ch | Kernel | Stride | Pad | Notes |
|---|---|---|---|---|---|---|
| Transposed Convolution | 32 | 128 | 4 | - | 1 | Batch Norm. |
| Transposed Convolution | 256 | 128 | 4 | - | 1 | Batch Norm. |
| Transposed Convolution | 256 | 64 | 4 | - | 1 | Batch Norm. |
| Transposed Convolution | 128 | 32 | 4 | - | 1 | Batch Norm. |
| Transposed Convolution | 64 | 16 | 4 | 2 | 1 | Batch Norm. |
| Transposed Convolution | 32 | 32 | 4 | 2 | 1 | Batch Norm. |
| Convolution | 32 | 32 | 5 | 1 | 2 | - |
| Convolution | 32 | 3 | 5 | 1 | 2 | RGB Output |

Table 6: Feature Decoder Architecture.

**Basis Modulation and Phase-PGF.** To combine the T-PNF and S-PGF discussed above, we introduce the procedure of basis modulation and form Phase-PGF to represent the dynamic scene.

Intuitively, the dimension of the phase space is determined by the number of subbands $\mathcal{K}$ in S-PGF. For each $g_j \in R^{(\infty)}(a, b, d(\theta_j), d(\mu_j), d(\gamma_j), \delta)^d$, we assign the phase $h_j(t)$ to it. Therefore, the subband $\mathcal{G}_j(\mathbf{x}, a, b)$ is modulated to:

$$\mathcal{G}'_j(\mathbf{x}, t, a, b) = \mathbf{W}_k g_j(\mathbf{x} + h_j(t)), g_j \in R^{(\infty)}(a, b, d(\theta_j), d(\mu_j), d(\gamma_j), \delta)^d, \mathbf{W}_k \in \mathbb{R}^{h \times d}. \quad (59)$$

Please refer to appendix C.2 for a more comprehensive discussion on this topic.

**Neural Rendering.** In the case of 3D dynamic scene, we further perform a neural rendering (Mildenhall et al., 2021) to render the 2D feature fields. The first 32 output dimension of Phase-PGF is interpreted as feature $\mathbf{f}(\mathbf{x})$ and the last dimension is interpreted as density $\sigma(\mathbf{x})$ used in neural rendering. The 2D feature map $m$ is rendered using the following volume rendering equation:

$$F(\mathbf{r}) = \int_{t_n}^{t_f} T(t)\sigma(\mathbf{r}(t))\mathbf{f}(\mathbf{r}(t), \mathbf{d}) \, \mathrm{d}t, \quad (60)$$

where $T(t) = \exp\left(-\int_{t_n}^{t} \sigma(\mathbf{r}(s)) \, \mathrm{d}s\right)$, $\mathbf{r}$ is the ray being rendered, and $d$ is the view direction.

**Feature Decoding and Rendering.** We then describe the network architecture that decodes a feature map into RGB rendering. The decoder is a bunch of U-Net (Ronneberger et al., 2015) like convolution layers followed by batch normalization. It incorporates skip connections for feature concatenation. Starting with 256 input channels, the decoder progressively reduces and then increases the channel size. The final layers produce outputs suitable for RGB images, with a Tanh activation function scaling the outputs to a specific range. The detailed architecture of it is detailed in Table 6.

The Upsampler module employs nearest neighbor upsampling and convolution operations for feature map upscaling. It includes two stages of upsampling, each magnifying the spatial dimensions by a factor of 2. Between these stages, convolutional layers adjust the channel dimensions. The configuration is outlined in Table 7.

**Adversarial Training.** During adversarial training, a PatchGAN (Isola et al., 2017) architecture discriminator is used.

| Operation | In Ch | Intermediate Ch | Out Ch | Notes |
|---|---|---|---|---|
| Upsample (x2) | - | - | - | Nearest Neighbor |
| Convolution | 32 | 128 | - | Kernel: 3, Pad: 1 |
| ReLU Activation | - | - | - | - |
| Upsample (x2) | - | - | - | Nearest Neighbor |
| Convolution | 128 | 3 | - | Kernel: 3, Pad: 1 |

Table 7: Upsampler Architecture.

## C.2 DISCUSSION ON THE PHASE SPACE

In this section, we further discuss the initialization and the structure of the phase space. We also perform an experiment to help the reader intuitively understand the phase space.

As previously discussed in appendix C.1, the phase space has a dimension $\mathcal{K}$, which is identical to the number of subbands defined in S-PGF. This parameter is further determined by the human prior knowledge of the number of motion components in the given scene. However, we argue that our architecture allows a redundant number of phase dimensions that can be not identical to the motion count in the given scene.

This mechanism of a redundant number of phase dimensions is similar to Slot-Attention (Locatello et al., 2020), where there can be empty slots that do not represent an object. In our case, whether a phase represents a part of motion information is determined by the *contribution* to the final rendering of the subband corresponding to this phase. Mathematically, this is formulated as a *phase score $s_j$* defined as below:

$$s_j = \mathbb{E}_{t,\mathbf{x}}||F_j(\mathbf{x}, t) - \mu_j(\mathbf{x}, t)||, \tag{61}$$
$$\mu_j(\mathbf{x}, t) = \mathbb{E}_{\mathbf{x}}F_j(\mathbf{x}, t) \tag{62}$$

Practically this score is calculated by performing discretization.

We provide an example below to demonstrate this argument. As demonstrated in Figure 7(a), we synthesize a video containing a ball with the motion of damping vibration. We instantiate our network with 16 subbands, each associated with a phase. Although such a phase space is redundant, the space can be analyzed using eq. (62). The result is shown in Figure 7(e), from which we can see that phase with index 0 and 2 dominated with score distribution.

We further sample some phases in visualize them in Figure 7(d). For phases having a high score (0 and 2), they align well with the ground truth motion shown in Figure 7(c). From the subband visualization in Figure 7(b), it can also be seen that they spatially represent the moving ball.

For phases that have a low score (4 and 9), they have a relatively chaotic motion. In Figure 7(b), it can be seen that they do not represent meaningful information in the input, indicating that they are "empty phases".

## C.3 MANIPULATION OF THE PHASE SPACE

In this section, we discuss the implementation details of the two different types of phase manipulations studied in this paper. The result of such manipulation is discussed in Sec. 4.3.

**Phase Smoothing.** The operation of phase smoothing corresponds to "motion smoothing" in Sec. 4.3. Practically, a bandwidth upper limit $B$ is defined to perform the editing. For a given phase sequence $h(t) \in \mathcal{H}$, the edited phase $h'$ is defined as follows:

$$h'(t|B) = \mathcal{F}^{-1}(T(\mathcal{F}(h(t))|B)), \tag{63}$$

$$T(f|B) = \begin{cases} 1, & \text{if } f \leq B, \\ 0, & \text{otherwise}, \end{cases} \tag{64}$$

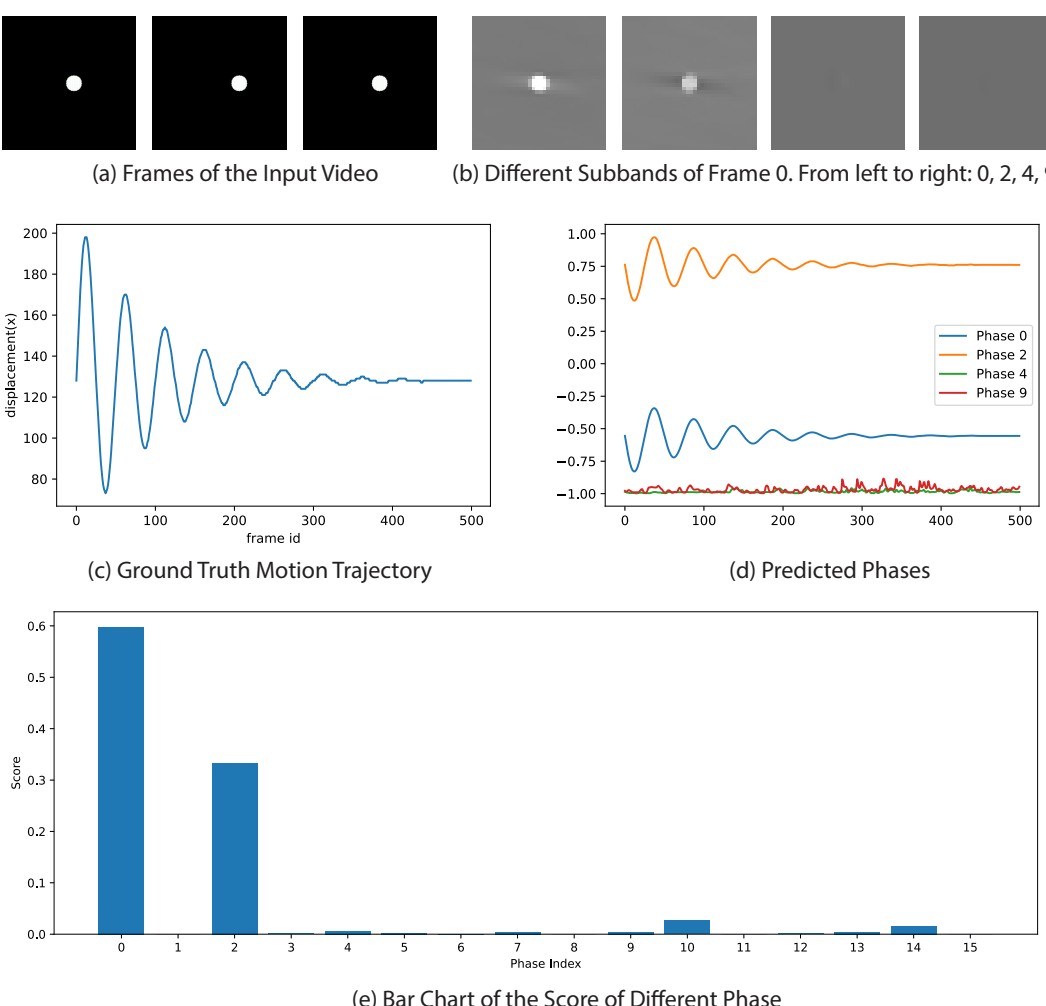

(a) Frames of the Input Video      (b) Different Subbands of Frame 0. From left to right: 0, 2, 4, 9

(c) Ground Truth Motion Trajectory      (d) Predicted Phases

(e) Bar Chart of the Score of Different Phase

Figure 7: Demonstration on having redundant phase space. (a) The input video shows a ball with a damping vibration motion. (b) Visualization of different subbands of the reconstructed model, evaluated at $t = 0$. Subbands 0 and 2 (corresponding to the phase 0 and 2) have motion information, while Subbands 4 and 9 (corresponding to the phase 4 and 9) are empty subbands. (c) The ground truth motion trajectory of a damping vibration. (d) Some of the predicted phases in the redundant phase space. Phases 0 and 2 were learned to represent motion information, while Phases 4 and 9 are empty. Please see `https://chen-geng.com/phasepgf#damping` for the animation.

where $\mathcal{F}$ and $\mathcal{F}^{-1}$ are Fourier transform and its inverse transform.

**Phase Intensity Adjustment.** This operation corresponds to "motion intensity adjustment" in Sec. 4.3. Practically, similar to the phase augmentation procedure described in Sec. 3.4, three parameters $\lambda, b_l, b_h$ are defined in this procedure. Given a phase sequence $h(t) \in \mathcal{H}$, the manipulated phase $h'$ is defined as:

$$h'(t|\lambda, b_l, b_h) = h(t) + (\lambda - 1) \cdot y(t|b_l, b_h), \tag{65}$$

$$y(t|b_l, b_h) = \mathcal{F}^{-1}(T(\mathcal{F}(h(t))|b_l, b_h)), \tag{66}$$

where $\lambda$ is intensity manipulation coefficient, $b_l$ and $b_h$ are subband limits for specific component of the signal, $\mathcal{F}$ and $\mathcal{F}^{-1}$ are Fourier transform and its inversion, and $T(f)$ is a band-limit filter defined as follows:

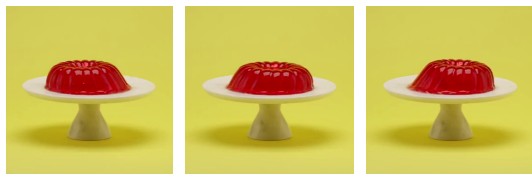

(a) Frames of the Input Video

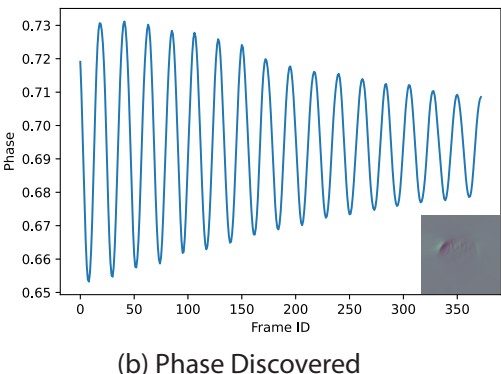

(b) Phase Discovered

Figure 8: Results on Non-Rigid Motion. In (a), we show the sampled frames of the input video. In (b), we show the discovered phase and associated subband. For animated video, please refer to `https://chen-geng.com/phasepgf#jelly`

$$T(f|f_1, f_2) = \begin{cases} 1, & \text{if } f_1 \leq f \leq f_2, \\ 0, & \text{otherwise.} \end{cases} \tag{67}$$

## C.4  TRAINING DETAILS

The models are trained on a NVIDIA A5000 GPU. The first stage of training takes around ten hours to converge. The second stage of adversarial training takes around three days to fully converge.

## D  ADDITIONAL EVALUATIONS

In this section, we provide more extensive results of the proposed method to further explore the boundary of the proposed method.

### D.1  MORE MACRO MOTION COMPLEXITIES

We first demonstrate several additional motion examples to show that our method can handle varying motion complexities.

**Non-Rigid Motion**   We first show that the proposed pipeline can also deal with non-rigid motion in a dynamic scene. Specifically, we run the proposed method on an Internet video dubbed *Jelly*. The results can be seen in Figure 8 and `https://chen-geng.com/phasepgf#jelly`.

It can be seen that although the motion in this video is a complex non-rigid, non-regular motion of a jelly, our method successfully discovered its low-dimension phase information by assigning a subband to its moving texture feature, as shown in Figure 8(b).

**Interaction Between Objects.**   A more complex motion pattern is when there are interactions between different objects. To simulate this case, we synthesize a video called *Collision* where there are two moving balls in the scene, and the first ball is collision with the second ball.

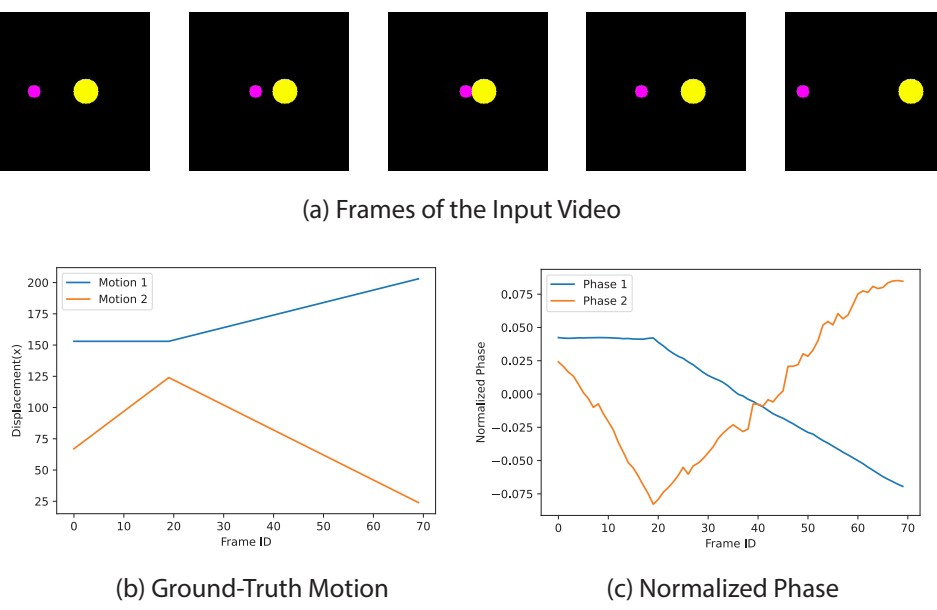

Figure 9: Results on Object Interaction. In (a), we show the sampled frames of the input video. We show the ground-truth motion trajectory in (b). In (c), we show the discovered phases (with normalization). For animated video, please refer to `https://chen-geng.com/phasepgf#collision`

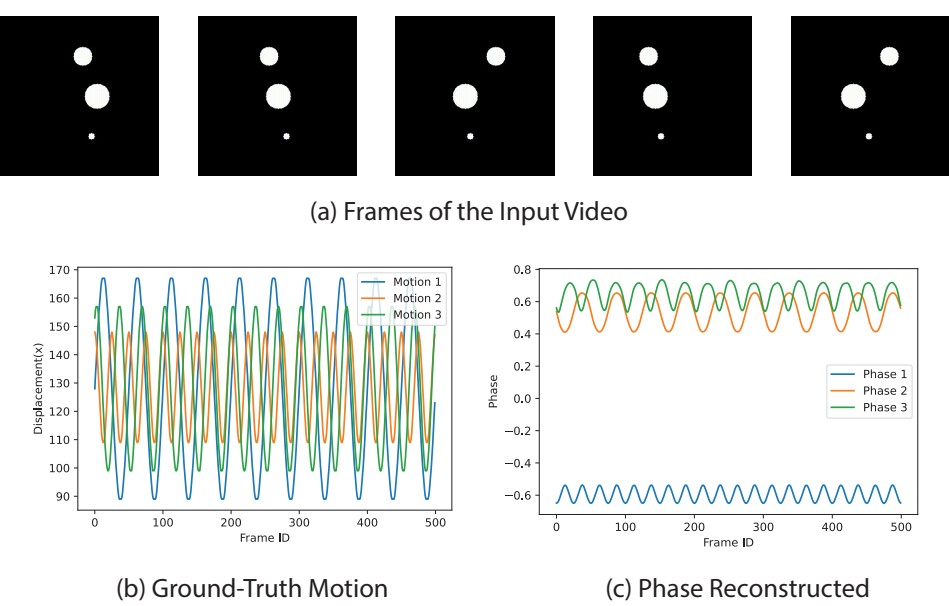

Figure 10: Results on Multi Objects Motion. In (a), we show the sampled frames of the input video. We show the ground-truth motion trajectory in (b). In (c), we show the discovered phases. For animated video, please refer to `https://chen-geng.com/phasepgf#ball3`

We show the results of this case in Figure 9 and `https://chen-geng.com/phasepgf#collision`. Our model can successfully decompose the phase space and discover plausible motion patterns: The first ball is moving initially and will change direction later, and the second ball is static at first while will be moving after the collision.

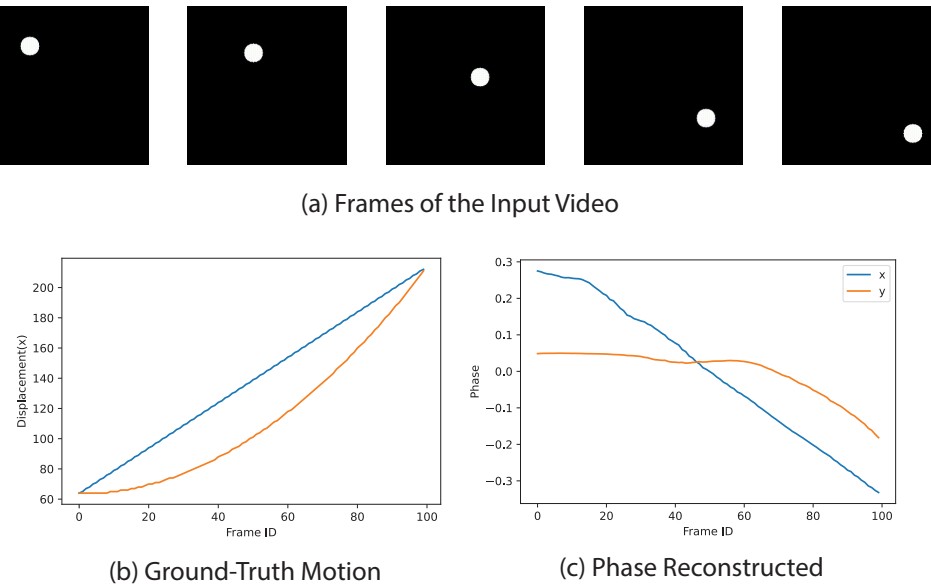

(a) Frames of the Input Video

(b) Ground-Truth Motion

(c) Phase Reconstructed

Figure 11: Results on Non-Periodic Motion. In (a), we show the sampled frames of the input video. We show the ground-truth motion trajectory in (b). In (c), we show the discovered phases. For animated video, please refer to `https://chen-geng.com/phasepgf#projectile`

**Multi-Object Motion.** We also study the applicability of the proposed framework in the scenario of a slightly more complex motion space. We synthesize a dynamic scene with three different moving balls (*Ball3* data) with different moving frequencies and trajectories. By using the proposed method to extract the phase information in such a scene, we can get the phase space decomposition as shown in Figure 10. The animated results can be found at `https://chen-geng.com/phasepgf#ball3`

**Non-Periodic Motion** We additionally synthesize three videos, *Damping*, *Bouncing*, and *Projectile*, containing different non-periodic motions. In *Damping* and *Bouncing*, we simulate damping vibration and bouncing of a ball. In *Projectile*, we simulate a ball being projected outward, subjecting to gravity.

The results in *Damping* can be found at Figure 7. It can be seen that the low-dimensional motion can be faithfully reconstructed by the extracted phase. Please see `https://chen-geng.com/phasepgf#damping` for the animation.

The *Projectile* example provides another different pattern of motion. From Figure 11, it can be observed that our phase, defined in two dimensions, can be used to decompose the given motion into x-dim and y-dim, representing uniform linear motion and parabolic motion, respectively. The animation can be found at `https://chen-geng.com/phasepgf#projectile`

## D.2 QUANTITATIVE METRICS AND EVALUATION

We introduce several metrics to quantitatively evaluate the results.

**Normalized Cross Correlation.** This metric is a measure used to quantify the similarity between two signals. For signal $f(t)$ and $g(t)$, this metric is defined as below:

$$r = \left| \frac{\sum_t (f(t) - \bar{f})(g(t) - \bar{g})}{\sqrt{\sum_t (f(t) - \bar{f})^2 \sum_t (g(t) - \bar{g})^2}} \right|, \tag{68}$$

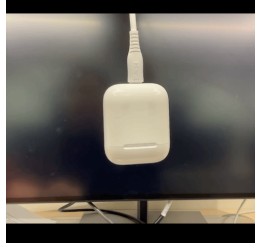 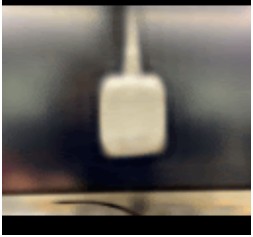 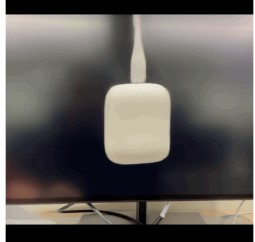 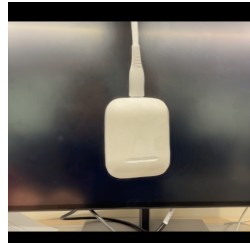

(a) Input      (b) Ours w/o Feat Dec. and Adv. Training      (c) Ours w/o Adv. Training      (d) Ours (Full Model)

Figure 12: Ablation Study. We study the effect of the proposed components by removing them from the full model. In (a), we show the input video. In (b), we show the rendering from the model without feature decoding and adversarial training. In (c), we show the rendering from the model without adversarial training. (d) shows the rendering of our full model.

We calculate this metric between the predicted phase and the ground truth motion trajectory in two synthetic data: *Projectile* and *Damping*. The results are shown in Table 2. Compared to the baseline, our model extracts phases that are more aligned to the input trajectory.

**Fréchet Inception Distance (FID).** We calculate the FID score (Heusel et al., 2017) between the input video and the manipulated video to evaluate the rendering quality of different methods. We perform the evaluation on a real captured video *Airpods*. The video can be found at `https://chen-geng.com/phasepgf`.

The result can be found at the third row of Table 3. Our method surpasses all the baselines significantly in terms of rendering quality.

## D.3 ABLATION STUDIES

We conduct ablation studies on the proposed components to validate the effectiveness of the proposed method. The results can be found at Figure 12.

**The Effect of Feature Decoder** It can be seen from Figure 12(b) and (c) that when we remove the feature decoding module, the rendering quality has degraded by a large magnitude. The reason is that only using S-PGF will produce low-resolution images. By using a feature decoder, we can render higher-resolution images.

**The Effect of Adversarial Training** By comparing Figure 12(c) and (d), we can see that adding adversarial training lets the model learn more information on the detailed texture of the object.

## D.4 MORE ANIMATED RESULTS

We refer the reader to the supplementary website: `https://chen-geng.com/phasepgf` for more animated results.

## D.5 DISCUSSION ON THE POINT TRACKING METHODS

Another possible method to perform macro motion analysis is by doing a dense point tracking method and figuring out some method to extract sparse motion information from the dense tracked particle trajectory. However, it is non-trivial to perform this process. After point tracking, the obtained particle trajectories are of a high dimension, which can not be easily analyzed.

We make an attempt in this section to use a State-of-the-Art point tracking method PIPs++ (Zheng et al., 2023) together with a dimension reduction technique to get a sparse motion feature sequence. Specifically, we sample dense tracked particles using a grid strategy following (Zheng et al., 2023) and perform dense point tracking. This results in a high-dimensional motion trajectory. We then

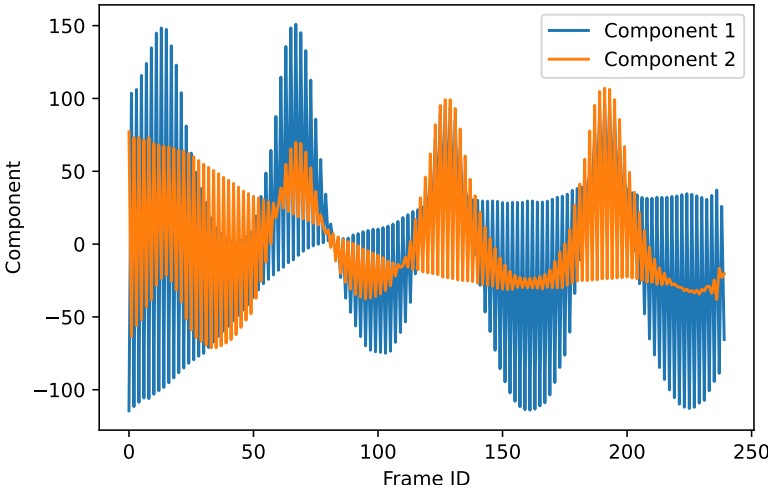

Figure 13: Two major components from the point tracking baseline. The phase generated by our method can be found at Figure 3. See visualization of tracked particles at `https://chen-geng.com/phasepgf/tracking.html`

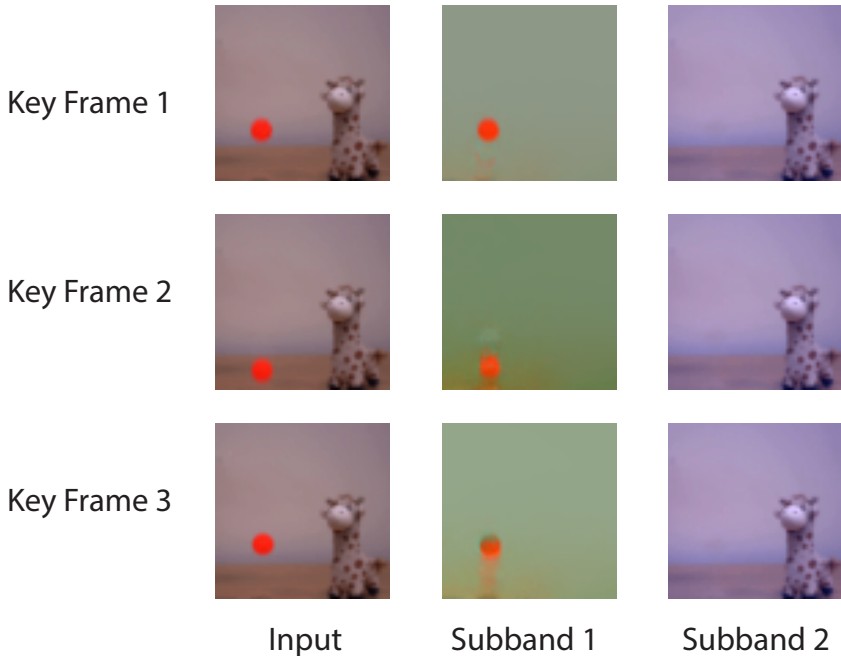

Figure 14: Moving object discovery on *Giraffe* data. By running the proposed method on the data, the moving object and static objects are separated into different subbands.

perform Principal Component Analysis (PCA) on the obtained dense motion to extract its dominant dimensions to get a sparse motion trajectory. The animated result can be found at `https://chen-geng.com/phasepgf//tracking.html`.

We visualize the extracted motion components in Figure 13. It can be seen that the extracted low-dimensional motion representation is noisy and is not plausible for the motion macro motion analysis task in this paper (Cf. the phase generated by the proposed method in Figure 3).

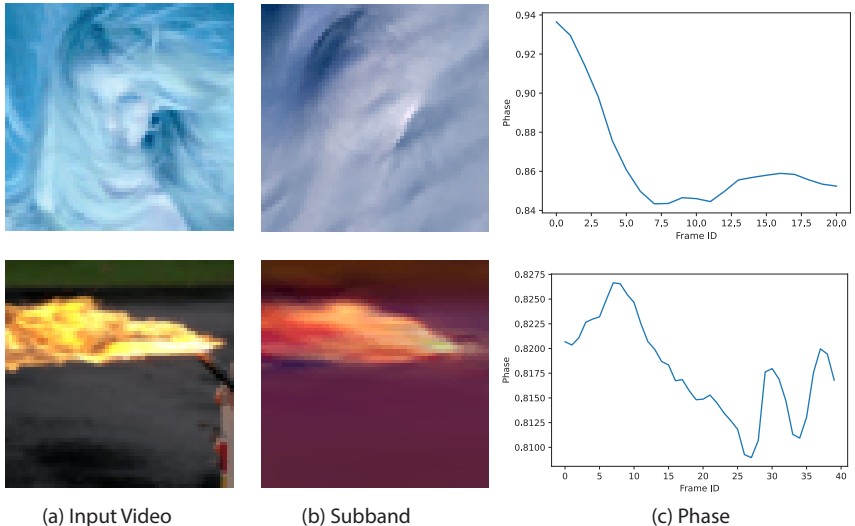

|         |         |         |
| (a) Input Video | (b) Subband | (c) Phase |

Figure 15: Failure cases. Animation: `https://chen-geng.com/phasepgf#failure`

## D.6    DISCUSSION ON MOVING OBJECT DISCOVERY

The proposed method can be potentially used for finding moving objects in a dynamic scene. In this section, we make a preliminary attempt at this.

We capture a video called *Giraffe* and run our method on this video. The result can be found in Figure 14. In this example, the moving red ball and the static giraffe toy are separated into two different subbands. Using this decomposition, we can detect the moving objects in a dynamic scene.

## D.7    FAILURE CASES

In this section, we provide some failure cases of the proposed method to help the reader better understand the boundaries and limitations of the proposed method. The results can be found in Figure 15 and please refer to `https://chen-geng.com/phasepgf#failure` for animation in this section.

**Chaotic Motion *Hair***    For chaotic motion such as hair blowing, there is no underlying low-dimensional representation of the scene motion. In this case, the method can only discover a coarse motion tendency yet cannot recover the full motion space.

**Motion of Shapeless Objects *Fire***    The proposed method assumes the objects in the scene have a concrete shape. For data like fire where the objects are shapeless, our method can only discover the very coarse moving part, but cannot clearly separate different motion parts.

