# OpenReview forum: "Neural Polynomial Gabor Fields for Macro Motion Analysis"
_ICLR.cc/2024/Conference — ICLR 2024 poster_

### Official Review · Reviewer_NjSx · 2023-10-29

**Soundness:** 3 good
**Presentation:** 3 good
**Contribution:** 3 good
**Rating:** 6
**Confidence:** 2

**Summary:**

This paper focuses on macro motion analysis, which involves collecting all visually observable motions in dynamic scenes. The traditional filtering-based methods typically pay attention to local and tiny motions, while recent dynamic neural representations can represent motions faithfully, but cannot be directly applied to motion analysis. The authors propose Phase-based neural polynomial Gabor fields (Phase-PGF) to represent scene dynamics with low-dimensional time-varying phases. Phase-PGF has several properties suitable for macro motion analysis, and it can be used for various dynamic scene editing tasks, such as motion loop detection, motion separation, motion smoothing, and motion magnification. It further implements the Phase-PGF using an innovative neural architecture and a refined training approach to enhance the quality of dynamic scene representation and editing.

The main contributions of this paper are as follows:
1) formulate the macro motion analysis problem
2) provides a novel phase-based neural polynomial Gabor fields (Phase-PGF) approach to tackle the motion analysis problem
3) demonstrates the effectiveness of the Phase-PGF approach on both 2D and 3D scenes

---------------------------------------------------------------------------------------------------------------------------------------
After reviewed the author's rebuttal, I have increased my score.

**Strengths:**

The main strengths of this paper are as follows:
1. Originality: The paper proposes a novel approach called Phase-based neural polynomial Gabor fields (Phase-PGF) for representing and analyzing dynamic scenes. This approach addresses the gap in the field of computer vision, where traditional methods focus on local and tiny motions, while recent dynamic neural representations lack direct applicability for macro motion analysis. The paper's originality lies in its focus on macro motion analysis and the introduction of Phase-PGF as a suitable representation for this purpose.
2. Quality: The paper presents a detailed analysis of the proposed approach, discussing its theoretical properties and demonstrating its ability to handle various macro motion analysis tasks such as motion separation, motion smoothing, and motion magnification. The experiments conducted showcase the effectiveness of Phase-PGF in representing and editing dynamic scenes, making the paper's contributions of high quality.
3. Clarity: The paper is well-structured and clear. The authors formulate the problem of macro motion analysis, explain the theoretical properties of Phase-PGF, and discuss the implementation and training of the approach. The experiments are described in sufficient detail, making it easy for readers to understand and replicate the proposed method.
4. Significance: The paper addresses a key challenge in the field of computer vision, motion analysis, and dynamic scene representation. By focusing on macro motion analysis, the proposed approach has the potential to impact various applications, such as motion tracking, generation, virtual reality, and other computer vision tasks. The paper's contributions are significant in providing a novel solution to representing and analyzing large motions and 3D scenes, which has been a largely underexplored area in the field.

In summary, the main strengths of the paper are its originality, quality, clarity, and significance. The paper presents a novel and effective approach to representing and analyzing macro motions in videos (both 2D and 3D), discusses its theoretical properties, and demonstrates its practical applicability through experiments, making it a valuable contribution to computer vision and related domains.

**Weaknesses:**

Weaknesses of the paper:
1. Slight artifacts in boldly magnifying large motions: The paper acknowledges that when magnifying large motions, there are slight artifacts in Phase-PGF. The cause of these artifacts may stem from the neural network architecture or the way motion is represented. To reduce these artifacts, additional investigation and optimization of the model may be necessary.

2. Limited scalability to complex large-scale 3D dynamic scenes: The paper acknowledges that Phase-PGF might not perform well in complex large-scale 3D scenes due to computational efficiency issues. As the paper's focus is on macro motion analysis, addressing this issue is crucial to improving the applicability of the proposed method, especially for complex scenes. A possible solution could be the use of a spatially adaptive Gabor basis as mentioned in the paper.

3. Insufficient experimental evaluation: Although the paper presents some experimental results, it would be beneficial to include more comprehensive evaluations with various datasets and tasks. For example, additional experiments could be conducted on larger and more diverse datasets, varying scene conditions, and different motion types (e.g. human motion). This would help establish the robustness and generalizability of the proposed method.

4. Neural network architecture: The paper uses a neural network to instantiate the Phase-PGF representation. However, the choice of the neural network architecture could be further optimized for the specific task of macro motion analysis. Including the details of the architecture may help to improve the reproducibility of the proposed work.

**Questions:**

1. In the related work section, the authors discuss concurrent works on tiny motion editing. How does Phase-PGF differ from these methods in terms of addressing macro motion analysis?

2. The authors mention that Phase-PGF may present slight artifacts when magnifying large motions. Could they provide further insights on the reasons for these artifacts and potential solutions to address this issue?

3. Are there plans to improve Phase-PGF's scalability for large-scale 3D dynamic scenes?

4. In the experimental section, it would be helpful to know more about the datasets used and the evaluation metrics employed for assessing the performance of Phase-PGF. For example, for the human preference study, how many ratings are obtained for each video, and how many videos are being used in the study?

5. Can the authors discuss the potential applications of Phase-PGF beyond the mentioned motion analysis and editing tasks, such as object tracking, motion generation, or other real-world scenarios?

---

> ### Author Response · Authors · 2023-11-22
> **Response to Reviewer NjSx (Part 1 / 2)**
>
> Thank you for your constructive review and insightful suggestions!
>
> We address all your concerns and questions as below.
>
> ### Addressing slight artifacts
> One possible reason for the slight artifacts in motion magnification is that the spatial support is insufficient for the magnified object trajectory. As we discussed in the limitation section, this might be addressed by adding more Gabor basis which requires more computational resources, or by introducing spatially-adaptive Gabor basis in future work.
>
> ### Plan for scaling to complex 3d dynamic scenes
> As dicussed in the limitation section, an essentialy bottleneck in scaling our method to large-scale complex 3D dynamic scene is adding more Gabor basis and subbands, which reuiqres increased compute. An alternative solution is spatially adaptive Gabor basis, yet we remain this for future work.
>
> ### Further experimental evaluation:
> We have included additional supporting evaluations in the revised manuscript appendix D, as detailed below:
>
> **Additional examples.** We include 6 more examples of more complex macro motions:
> - **Non-rigid motion**: We include an Internet video, *Jelly*, where a jelly is wobbling on a seat. This example demonstrates non-rigid non-uniform shearing motion.
> - **Interaction between objects**: We include an example of two balls colliding with each other and then changing moving directions.
> - **Multi-object motion**: We include a scene with three balls vibrating with different frequencies and magnitudes.
> - **Non-periodic motion**: We synthesize three videos, *Damping*, *Bouncing*, and *Projectile*, contain different non-periodic motions. In *Damping* and *Bouncing*, we simulate a damping motion of a ball. In projectile, we simulate a ball being projected outward, subjecting to gravity.
>
> We show further descriptions, results, and discussion in Appendix D.1. In summary, our model is able to represent and interpret the more diverse motions in these additional examples. Video results are also available at our [supplementary webpage](https://phasepgf.github.io/more_results.html).
>
> **Quantitative results.** We also include two quantitative metrics to evaluate all the examples, including:
> - Normalized Cross-Correlation (NCC) to measure for phase correlation and phase interpretability. The NCC is a standard metric in time-series analysis to measure the similarity of two 1-D temporal signals, which meet our need in measuring how the generated phase align with the motion is a dynamic scene. In Table 2, we show that our method surpasses the baseline significantly in this metric.
> - Frechet Inception Distance (FID) to measure the quality of the phase-manipulated videos. In particular, we compute FID between the manipulated video and the original video. Since our motion manipulation only changes the motions presented in the scene, a lower FID then suggests less artifacts and higher visual fidelity. In Table 3, we show that the result produced by our method has better visual quality.
>
> We include more disucssion on quantitative metrics in Appendix D.2.
>
> **Phase space analysis and moving object discovery.** To provide more analysis, we also add an analysis on phase space. In particular, we introduce a quantity, *phase score*, to analyze and automatically select phases that correspond to significant motions in the dynamic scene. We discuss phase space in Appendix C.2. We show that our model can discover moving objects in a real video in Appendix D.6.
>
> **Ablation study.** In Appendix D.3, we include ablation study on decoder and adversarial training. In Figure 12 of the revised manuscript, we show that the introduced components contribute significantly to our model's high visual fidelity.
>
> **Comparison to point tracking-based method.** We include one additional point tracking-based baseline method. Specifically, we extract points using PIPs++ [1], a state-of-the-art point tracking method. To analysis motion, we use PCA to reduce its dimensions. We show a qualitative result in Figure 13. We observe that the motion space from point tracking is not structured or interpretable, and thus it does not support intrepreting or manipulating motions in a meaningful way. This suggests that the unstructured point-space representation does not easily support abstracting a low-dimensional manipulatable motion representation.
>
> We believe these additional experiments can better support our method.
>
> ### Network architecture:
> In Appendix C.1, We include all our additional technical details, including
> - Phase Generator
> - Spatial Polynomial Gabor Fields
> - Basis Modulation and Phase-PGF
> - Neural Rendering
> - Feature Decoding and Rendering
> - Adversarial Training
>
> We also added Table 4,5,6,7 that further specify the hyper-parameters and network architectures we used.

---

> ### Author Response · Authors · 2023-11-22
> **Response to Reviewer NjSx (Part 2 / 2)**
>
> (cont'd)
>
> ### Related work
>
> Phase-PGF introduces a completely different motion analysis framework to tackle the macro motion analysis task. Specifically, we use phase to capture the macro motion information, differing from other works significantly.
>
> ### Reproducibility
> We will release code and data upon acceptance.
>
> ### Details on human preference evaluation
> As indicated in Appendix B and the supplementary webpage, for human evaluation, over 100 ratings are obtained for each video. As indicated, please see our [supplementary webpage](https://phasepgf.github.io/) for the actual user study form we used.
>
> ### Further applications
> We demonstrate moving object discovery for video analysis. To automatically discover moving objects, we introduce a quantity, *phase score*, which quantifies how much a subband contributes to the final rendering. The subbands associated with the highest phase scores represent the dynamic/static objects in the scene. We discuss the automatic phase association in Appendix C.2. We show that our model can discover moving objects in a real video in Appendix D.6.

---

> > ### Comment · Reviewer_NjSx · 2023-11-22
> > **Thank you for the comprehensive rebuttal**
> >
> > I thank the authors for their comprehensive rebuttal. I believe the authors have addressed most of my concerns.

---

### Official Review · Reviewer_WSr6 · 2023-10-30

**Soundness:** 3 good
**Presentation:** 3 good
**Contribution:** 3 good
**Rating:** 6
**Confidence:** 4

**Summary:**

This paper formulates and studies a new task of macro motion analysis. To perform macro motion analysis, the authors propose to learn an implicit function with respect to point coordinate x and time t, composed of Gabor functions and phase functions, namely Phase-PGF. By studying and adjusting the phase function of the Phase-PGF, one can infer periodic motion detection, motion separation, motion smoothing and motion intensity adjustment. Additionally, in order for the Phase-PGF to be able to extrapolate to unseen motions, which is needed for motion intensity adjustment, a discriminator on decoded images is used for adversarial training. Experiments on 2D video and 3D dynamic scenes are performed. Both human preference and visual results show its effectiveness and correctness on several motion analyses tasks.

**Strengths:**

1. Overall, a new problem of macro motion analysis is properly formulated and approached with reasonable method, proven effective by adequate experiments.
2. Using phase functions to model dynamic scenes has advantages for motion analysis compared with other dynamic scene representations.
3. I appreciate authors experimenting with some real-captured data and the efforts of increasing image rendering quality.

**Weaknesses:**

1. It would be better to design some automatic metrics other than human preference. For example, trackers can be put on moving objects to obtain ground truth motions, which can be used to compare with predicted motions. For the motion intensity adjustment experiment, to evaluate the visual quality, some metrics designed for generation tasks, e.g., FID, KID, can be used.
2. Currently, the examples in paper and supplementary website show very simple movements, mostly periodic. I wonder how would the method apply to more complex, dynamic scenes. Not necessarily dynamic 3D scenes, more complex 2D video can also do the work. If so, can the method separate more than two motions in the scene? And can some algorithm be designed to automatically find the moving object?

**Questions:**

1. Currently, the extracted motion phase is agnostic of the motion direction. Is there a way to decompose motion phases, for example, to x and y directions.
2. How, in practice, are the motion representation, e.g., Figure. 2(a), extracted from learned implicit function?

---

> ### Author Response · Authors · 2023-11-22
> **Response to Reviewer WSr6**
>
> Thank you for your constructive review and insightful suggestions!
>
> We address all your concerns and questions as below.
>
> ### Quantitative metrics
> Following your suggestion, we use quantitative metrics to evaluate phase interpretability and video quality. In particular, we use:
> - Normalized Cross-Correlation (NCC) to measure for phase correlation and phase interpretability. The NCC is a standard metric in time-series analysis to measure the similarity of two 1-D temporal signals, which meet our need in measuring how the generated phase align with the motion is a dynamic scene. In Table 2, we show that our method surpasses the baseline significantly in this metric.
> - Frechet Inception Distance (FID) to measure the quality of the phase-manipulated videos. In particular, we compute FID between the manipulated video and the original video. Since our motion manipulation only changes the motions presented in the scene, a lower FID then suggests less artifacts and higher visual fidelity. In Table 3, we show that the result produced by our method has better visual quality.
>
> We include more disucssion on quantitative metrics in Appendix D.2.
>
> ### More complex motions
> We include 6 more examples of more complex macro motions:
> - **Non-rigid motion**: We include an Internet video, *Jelly*, where a jelly is wobbling on a seat. This example demonstrates non-rigid non-uniform shearing motion.
> - **Interaction between objects**: We include an example of two balls colliding with each other and then changing moving directions.
> - **Multi-object motion**: We include a scene with three balls vibrating with different frequencies and magnitudes.
> - **Non-periodic motion**: We synthesize three videos, *Damping*, *Bouncing*, and *Projectile*, contain different non-periodic motions. In *Damping* and *Bouncing*, we simulate a damping motion of a ball. In projectile, we simulate a ball being projected outward, subjecting to gravity.
>
> We show further descriptions, results, and discussion in Appendix D.1. In summary, our model is able to represent and interpret the more diverse motions in these additional examples. Video results are also available at our [supplementary webpage](https://phasepgf.github.io/more_results.html).
>
> ### Separate more than two motions
> In the Multi-object motion example above, we demonstrate separation of three balls.
>
> ### Automatic moving object discovery
>
> To automatically discover moving objects, we introduce a quantity, *phase score*, which quantifies how much a subband contributes to the final rendering. The subbands associated with the highest phase scores represent the dynamic/static objects in the scene. We discuss the automatic phase association in Appendix C.2. We show that our model can discover moving objects in a real video in Appendix D.6.
>
> ### Clarification on representing motion direction
> We clarify that our method does represent motion direction by the orientation of the Gabor basis. In particular, we do decompose motion directions into $x$ and $y$. This can be seen intuitively from the projectile example in Figure 11.
>
> ### Clarification on extracting motion representation
> We set small timesteps to sample the implicit function values and discretize the continuous implicit function to a discrete phase sequence. Specifically, we use a timestep $2/N$ in a normalized scale, where $N$ is the number of input frames.

---

### Official Review · Reviewer_no8y · 2023-10-31

**Soundness:** 2 fair
**Presentation:** 2 fair
**Contribution:** 2 fair
**Rating:** 6
**Confidence:** 2

**Summary:**

The paper proposes to represent scene dynamics with low-dimensional time-varying phases. The representation is learned using Phase-based neural polynomial Gabor fields (Phase-PGF). The paper argues that Phase-PGF has several properties which make it suitable for motion analysis task like motion loop detection, motion factorization,  and motion magnification.

**Strengths:**

- The paper addresses an interesting application and is nicely written and easy to read.
- The paper proposes a novel formulation (Phase-PGF) for scene motion.
- Phase-PGF several properties which make it suitable for motion analysis. Two properties I found interesting are :Periodicity correlation and motion separation.
- The paper show several interesting motion analysis applications like motion separation, motion factorization, motion magnifications
- The results shown seems to surpass previous work

**Weaknesses:**

- Although the method is novel, I think the supporting experiments are insufficient to validate the method.
- Not enough results in the supplementary webpage. All the results, except one, are visualizing the phase. Only one result show a generated video. This is insufficient to judge the quality of the results. I would have liked to see a video of the motion separation experiment and more videos for the motion intensity adjustment.
- Motion editing results display visual artifact
- Most of the experiments are on toy examples with simple objects or simple motion.

**Questions:**

- sometimes it is hard to say which phase is right. Maybe participants are just biased towards more periodic-looking signal even if it does not really match the scene motion. I wonder if there is more systematic way to evaluate this? For example, we can detect the periodicity of a given motion and then compare its frequency with the predicted frequency.

---

> ### Author Response · Authors · 2023-11-22
> **Response to Reviewer no8y**
>
> Thank you for your constructive review and insightful suggestions!
>
> We address all your concerns and questions as below.
>
> ### Additional supporting experiments
> To include additional supporting experiments, we extensively include additional contents.
>
> **Additional examples.** We include 6 more examples of more complex macro motions:
> - **Non-rigid motion**: We include an Internet video, *Jelly*, where a jelly is wobbling on a seat. This example demonstrates non-rigid non-uniform shearing motion.
> - **Interaction between objects**: We include an example of two balls colliding with each other and then changing moving directions.
> - **Multi-object motion**: We include a scene with three balls vibrating with different frequencies and magnitudes.
> - **Non-periodic motion**: We synthesize three videos, *Damping*, *Bouncing*, and *Projectile*, contain different non-periodic motions. In *Damping* and *Bouncing*, we simulate a damping motion of a ball. In projectile, we simulate a ball being projected outward, subjecting to gravity.
>
> We show further descriptions, results, and discussion in Appendix D.1. In summary, our model is able to represent and interpret the more diverse motions in these additional examples. Video results are also available at our [supplementary webpage](https://phasepgf.github.io/more_results.html).
>
> **Quantitative results.** We also include two quantitative metrics to evaluate all the examples, including:
> - Normalized Cross-Correlation (NCC) to measure for phase correlation and phase interpretability. The NCC is a standard metric in time-series analysis to measure the similarity of two 1-D temporal signals, which meet our need in measuring how the generated phase align with the motion is a dynamic scene. In Table 2, we show that our method surpasses the baseline significantly in this metric.
> - Frechet Inception Distance (FID) to measure the quality of the phase-manipulated videos. In particular, we compute FID between the manipulated video and the original video. Since our motion manipulation only changes the motions presented in the scene, a lower FID then suggests less artifacts and higher visual fidelity. In Table 3, we show that the result produced by our method has better visual quality.
>
> We include more disucssion on quantitative metrics in Appendix D.2.
>
> **Phase space analysis and moving object discovery.** To provide more analysis, we add an analysis on phase space in Appendix C.2. Following the analysis on phase space, we introduce a *phase score* metric to automatically discover the associated subbands. The phase score quantifies how much a subband contributes to the final rendering. The subbands associated with the highest phase scores represent the dynamic/static components in the scene. We show that our model can discover moving objects in a real video in Appendix D.6.
>
> **Ablation study.** In Appendix D.3, we include ablation study on decoder and adversarial training. In Figure 12 of the revised manuscript, we show that the introduced components contribute significantly to our model's high visual fidelity.
>
> **Comparison to point tracking-based method.** We include one additional point tracking-based baseline method. Specifically, we extract points using PIPs++ [1], a state-of-the-art point tracking method. To analysis motion, we use PCA to reduce its dimensions. We show a qualitative result in Figure 13. We observe that the motion space from point tracking is not structured or interpretable, and thus it does not support intrepreting or manipulating motions in a meaningful way. This suggests that the unstructured point-space representation does not easily support abstracting a low-dimensional manipulatable motion representation.
>
> We believe these additional experiments can better support our method.
>
> ### More video results on webpage
> We add video visualizations for video results for motion analysis, motion smoothing, motion magnification, motion separation, and failure cases in our [supplementary webpage](https://phasepgf.github.io/more_results.html).
>
> ### Quality for video results
> As summarized above, we add a quantitative metric FID for video quality and show results in Table 8. We achieve higher quality compared to baseline methods.
>
> ### More complex motions
> As summarized above, we add several examples in more complex motions. Among them, the *Jelly* example demonstrates not only more complex motion that includes non-rigid shearing, but also more complex object, i.e., a colorful soft body. We show that our model can reconstruct well the video and generate an interpretable motion representation including a dominant phase in Figure 8.
>
> ### Systematic evaluation on phases
> As summarized above, we use a metric, Normalized Cross Correlation, to measure phase correctness. Please refer to appendix D.2 for more details.
>
> [1] Zheng et al., PointOdyssey: A Large-Scale Synthetic Dataset for Long-Term Point Tracking, ICCV, 2023

---

### Official Review · Reviewer_EQJi · 2023-11-01

**Soundness:** 3 good
**Presentation:** 3 good
**Contribution:** 3 good
**Rating:** 6
**Confidence:** 4

**Summary:**

This paper delves into the exploration of motion analysis through the use of phase-based representation, with an emphasis on large-scale, rigid-body movements, also known as macro motion. In contrast to localized and small-scale motion, macro motion operates on a broader scale, encompassing extensive spatial and temporal transitions that are easily perceptible by humans. Traditional methods often fall short in accurately capturing such features because the filter operations are primarily designed to handle low-level features.

To overcome this limitation, this paper introduces a novel approach that utilizes phase-oriented neural Gabor fields as input embedding to reconstruct the original images or 3D scenes. By a reverse optimization process, the optimized phase input can encode some useful information while keeping the periodic. As the fundamental element in the real world, some good properties of phase can be also useful in further analysis. To enhance this reconstruction and reverse optimization, various training strategies like multi-stage, deep latent, and adversarial training have been proposed. Despite the lack of extensive evaluation, these methods are expected to deliver promising results.

In the experimental section, through some examples, it shows the method can outperform some baseline methods in extracting more reliable alignment. Moreover, the properties of the phase support the smoothing and separation of motions, which can be beneficial in specific applications like magnification.

**Strengths:**

The story is clear, the reader can easily understand the importance of the phase and the effectiveness of the proposed method.

The idea is interesting and novel. Using the phase as the fundamental feature to describe the motion can help to understand the motion better.

I do like the inverse optimization with the Gabor field for locating the best-aligned phase feature; the theoretical exposition is clear, straightforward, and logical; and the proofs are also clear.

The employed coarse-to-fine generation framework and strategy effectively facilitate the learning of priors.
The results provide some evidence to support the central thesis of the paper.

**Weaknesses:**

While I appreciate the innovative concept presented in this paper, I believe the current version exhibits some weaknesses that need to be addressed:

Some of the exposition needs to be improved. Some details and properties are not clearly clarified. Some points need more in-depth discussion:

1. The input phase space and the details of the Phase Generator are not elaborated. As the central element in this model, it’s not clear how to initialize the number of phases, and how the current framework adapts to the diverse scene. The full phase space can be much more complex than that in those demonstrated examples. Hence the potential of this method is unclear.

2. The manipulation of the phase has been mentioned a few times, but it hasn't been detailed and demonstrated further in the paper. If the phase space is manipulatable, it will be important in the model interpretability.

3. The implementation of using Phase-PGF in the deep neural network is not described clearly enough.

The contribution is unclear with insufficient evaluation. A more comprehensive evaluation is needed to demonstrate the boundaries of working examples, and provide more insights:
1. Quantitative evaluation: Some of the test videos are artificial. In this case, we may want to synthesize more videos that contain varying motion complexities, for a more extensive evaluation.
2. Ablation study: An ablation study for each network module would be beneficial.
3. Evaluation metrics: A metric to measure if the phase meets the requirements would be useful.

Last, some existing methods use different formulations to perform similar motion analysis, it is strongly recommended to add the comparison with them. For example, some point tracking methods can also extract rigid body movements as sparse points, and using these predictions it's also possible to extract interpretable features of the motion, such as phase or other high-level descriptions. While I don't question the novelty of this paper as it employs a *unique* methodology, it would be beneficial to add more experiments and discussions on these approaches.

**Questions:**

Where is the boundary of the phase space this method can support? There can be highly varied frequencies in the real world, but the working range of the proposed method is unclear. It would be useful to explore these boundaries by conducting experiments with various artificial videos.

How to determine the number of phases? In some cases, there appears to be one phase, while in others there are multiple. Is it possible to use an excessive number of phases to overfit a scene? Conversely, what would be the outcome if we used a single phase for a scene with multiple objects?

The extracted phases seem to care about the frequency more compared to the amplitude variations. In the second video of Fig.2, different balls do have different spatial transitions, but they seem to share the same peak.

There is an option to use point tracking to extract the phase from the tracking path. How does this method complement them, and in what ways does it offer unique advantages compared to other methods?

Are the extracted phases open to manipulation? For example, if we multiply the phase by a coefficient A, will the corresponding recovered image or video be amplified? It is claimed in the paper that we can manipulate, but the results seem to be missing.

---

> ### Author Response · Authors · 2023-11-22
> **Response to Reviewer EQJi (Part 1 / 2)**
>
> Thank you for your constructive review and insightful suggestions!
>
> We address all your concerns and questions as below.
>
> ### Exposition on phase space and phase generator
>
> We add detailed exposition on the formulation and architecture of the Phase Generator in Appendix C.1 of the revised manuscript, and detailed discussion on phase space in C.2. In short, we have $K$ phase sequences, i.e., our phase space is $K$-dimensional, where $K$ denotes the number of subbands. Each of the $K$ generated phase sequences modulates a subband (we have also added details of the modulation in Appendix C.1). All $K$ subbands add up to collectively represent the dynamic scene.
>
> ### How to initialize the number of phases
>
> We add a detailed discussion in Appendix C.2 of the revised manuscript. The main idea is that, $K$ can be set either according to prior knowledge of how many motion components there are in the scene, or to a large number. When we set a large $K$, our model can automatically associate appropriate subbands (i.e., phases) to the motion components in the scene, thus automatically discovering moving objects without manually specifying the number of phases. Then, all other redundent phases have little or no effects on representing the dynamic contents.
>
> To do this, we introduce a *phase score* metric to automatically discover the associated subbands. The phase score quantifies how much a subband contributes to the final rendering. The subbands associated with the highest phase scores represent the dynamic objects in the scene. We show that our model can discover moving objects in a real video in Appendix D.6.
>
> ### Clarification on manipulating phases
>
> We clarify that several of our experiments on macro motion analysis indeed manipulate phases to manipulate motions. In particular:
> - **Motion separation (Figure 4)**: We used band-limit filters on the generated phase sequence, and then we isolated the motion of interest (the motion of the blue ball) according to the frequency. In this way, we manipulated (mollified) the motion of the blue ball without affecting the motion of the red ball.
> - **Motion intensity adjustment (Figure 5)**: We introduced a phase augmentation technique in Section 3.2., Equation (5) and (6), to magnify the motion in the video.
> - **Motion smoothing (Figure 6)**: We show that by removing high-frequency bands of the generated phase sequence, we can smooth out the high-frequency motions in the macro motion.
>
>
> We also include video results of phase-based manipulation in our [supplementary webpage](https://phasepgf.github.io/more_results.html).
>
>
>
> ### Details on implementation
> In Appendix C.1, We include all our additional technical details, including
> - Phase Generator
> - Spatial Polynomial Gabor Fields
> - Basis Modulation and Phase-PGF
> - Neural Rendering
> - Feature Decoding and Rendering
> - Adversarial Training
>
> We also added Table 4,5,6,7 that further specify the hyper-parameters and network architectures we used.
>
>
> ### Adapting to diverse motion complexities
>
> We include 6 more examples of more complex macro motions:
> - **Non-rigid motion**: We include an Internet video, *Jelly*, where a jelly is wobbling on a seat. This example demonstrates non-rigid non-uniform shearing motion.
> - **Interaction between objects**: We include an example of two balls colliding with each other and then changing moving directions.
> - **Multi-object motion**: We include a scene with three balls vibrating with different frequencies and magnitudes.
> - **Non-periodic motion**: We synthesize three videos, *Damping*, *Bouncing*, and *Projectile*, contain different non-periodic motions. In *Damping* and *Bouncing*, we simulate a damping motion of a ball. In projectile, we simulate a ball being projected outward, subjecting to gravity.
>
> We show further descriptions, results, and discussion in Appendix D.1. In summary, our model is able to represent and interpret the more diverse motions in these additional examples. Video results are also available at our [supplementary webpage](https://phasepgf.github.io/more_results.html).
>
>
> ### Ablation study
>
> In Appendix D.3, we include ablation study on decoder and adversarial training. In Figure 12 of the revised manuscript, we show that the introduced components contribute significantly to our model's high visual fidelity.
>
> (cont'd in the next response)

---

> ### Author Response · Authors · 2023-11-22
> **Response to Reviewer EQJi (Part 2 / 2)**
>
> ### Evaluation metrics
>
> To allow automatic quantitative measures, we additionally include the following evaluation metrics:
> - Normalized Cross-Correlation (NCC) to measure for phase correlation and phase interpretability. The NCC is a standard metric in time-series analysis to measure the similarity of two 1-D temporal signals, which meet our need in measuring how the generated phase align with the motion is a dynamic scene. In Table 2, we show that our method surpasses the baseline significantly in this metric.
> - Frechet Inception Distance (FID) to measure the quality of the phase-manipulated videos. In particular, we compute FID between the manipulated video and the original video. Since our motion manipulation only changes the motions presented in the scene, a lower FID then suggests less artifacts and higher visual fidelity. In Table 3, we show that the result produced by our method has better visual quality.
>
> We include more disucssion on quantitative metrics in Appendix D.2.
>
>
> ### More extenisve quantitative evaluation
> As mentioned above, we add quantitative measurements and additional examples in Appendix D of the revised manuscript. We believe these additional examples and evaluations add to the extensiveness of our experiments.
>
> ### Comparison to point tracking
> We follow your suggestion to compare with a point tracking-based method. In particular, we extract points using PIPs++ [1], a state-of-the-art point tracking method. To analyze motion, we use PCA to reduce its dimensions. We show a qualitative result in Figure 13. We observe that the motion space from point tracking is not structured nor interpretable, and thus it does not support intrepreting or manipulating motions in a meaningful way. This suggests that the unstructured point-space representation does not easily support abstracting a low-dimensional manipulatable motion representation.
>
>
> ### The working range of the proposed method
> Besides the five additional examples we have shown above, we also add failure cases to demonstrate the working range. In particular, we find that our approach does not work with motions of shapeless objects such as fire (Figure 15), because it violates our assumption on time-invariant basis functions to represent the video appearances. Another example is chaotic unpredictable motions. We show a hair example in Figure 15 where the hair motion is unpredictable and it might not allow a frequency-based low-dimensional representation.
>
>
> ### Amplitude of phases
> We clarify that the motion amplitudes can be captured by the generated phase amplitudes within **a single motion**. This can be observed from the *Damping* example in Figure 7.
>
> Yet, we note that the phase amplitudes are not comparable among different motions. For a scene with multiple motions, such as the two bouncing balls, although each ball's motion amplitude can still be captured, the relative phase amplitudes of the two balls do not reflect their relative motion amplitudes, because each phase signal is generated from an implicit function (phase generator), and an implicit function can have an arbitrary output scale.
>
> [1] Zheng et al., PointOdyssey: A Large-Scale Synthetic Dataset for Long-Term Point Tracking, ICCV, 2023

---

> ### Author Response · Authors · 2023-11-23
> **Kind Reminder on the Deadline of Discussion Period**
>
> Dear Reviewer,
>
> Thank you again for your insightful review and constructive feedback on our submission. We hope our responses have effectively addressed all your concerns. Please be informed that the author-reviewer discussion period ends today. Should there be any unresolved issues, we would appreciate your prompt feedback. If your concerns have been addressed, please kindly consider updating your score. Thank you!
>
> Best,
> Authors

---

> > ### Comment · Reviewer_EQJi · 2023-11-23
> > **Response to the authors**
> >
> > Thanks for the rebuttal, the additional visual results also did a nice job of clarifying. I believe most of my concerns have been addressed, increasing the rate to 6.

---

### Author Response · Authors · 2023-11-22
**Summary of Changes (Part 1 / 2)**

We thank all the reviewers for their insightful feedback. We have revised our manuscript accordingly and denote all changes by blue fonts. The main revisions are detailed below:

### [EQJi,no8y,WSr6,NjSx] Additional examples on more complex motions
We include 6 more examples of more complex macro motions:
- **Non-rigid motion**: We include an Internet video, *Jelly*, where a jelly is wobbling on a seat. This example demonstrates non-rigid non-uniform shearing motion.
- **Interaction between objects**: We include an example of two balls colliding with each other and then changing moving directions.
- **Multi-object motion**: We include a scene with three balls vibrating with different frequencies and magnitudes.
- **Non-periodic motion**: We synthesize three videos, *Damping*, *Bouncing*, and *Projectile*, contain different non-periodic motions. In *Damping* and *Bouncing*, we simulate a damping motion of a ball. In projectile, we simulate a ball being projected outward, subjecting to gravity.

We show further descriptions, results, and discussion in Appendix D.1. In summary, our model is able to represent and interpret the more diverse motions in these additional examples. Video results are also available at our [supplementary webpage](https://phasepgf.github.io/more_results.html).

### [EQJi] Analysis on phase space
We add a detailed discussion in Appendix C.2 on the phase space. The main idea is that, the dimension of the phase space $K$ is flexible and it can adapt to the dynamic scene. When we set a large $K$, our model can automatically associate appropriate subbands (i.e., phases) to the motion components in the scene, thus automatically discovering moving objects without manually specifying the number of phases. Then, all other redundent phases have little or no effects on representing the dynamic contents.

### [EQJi,no8y,WSr6,NjSx] Additional application: Moving object discovery

Following the analysis on phase space, we introduce a *phase score* metric to automatically discover the associated subbands. The phase score quantifies how much a subband contributes to the final rendering. The subbands associated with the highest phase scores represent the dynamic/static components in the scene. We show that our model can discover moving objects in a real video in Appendix D.6.


### [EQJi,NjSx] Implementation details

In Appendix C.1, We include all our additional technical details, including
- Phase Generator
- Spatial Polynomial Gabor Fields
- Basis Modulation and Phase-PGF
- Neural Rendering
- Feature Decoding and Rendering
- Adversarial Training

We also added Table 4,5,6,7 that further specify the hyper-parameters and network architectures we used.

### [EQJi,no8y,WSr6,NjSx] Quantitative metrics

To allow automatic quantitative measures, we additionally include the following evaluation metrics:
- Normalized Cross-Correlation (NCC) to measure for phase correlation and phase interpretability. The NCC is a standard metric in time-series analysis to measure the similarity of two 1-D temporal signals, which meet our need in measuring how the generated phase align with the motion is a dynamic scene. In Table 2, we show that our method surpasses the baseline significantly in this metric.
- Frechet Inception Distance (FID) to measure the quality of the phase-manipulated videos. In particular, we compute FID between the manipulated video and the original video. Since our motion manipulation only changes the motions presented in the scene, a lower FID then suggests less artifacts and higher visual fidelity. In Table 3, we show that the result produced by our method has lower FID than baseline methods.

We include more disucssion on quantitative metrics in Appendix D.2.

### [EQJi,no8y,NjSx] Ablation study

In Appendix D.3, we include ablation study on decoder and adversarial training. In Figure 12 of the revised manuscript, we show that the introduced components contribute significantly to our model's high visual fidelity.

(cont'd in the next global response)

---

### Author Response · Authors · 2023-11-22
**Summary of Changes (Part 2 / 2)**

(cont'd)

### [EQJi] Comparison to point tracking-based method

In Appendix D.5, we include a point tracking-based method and discuss the lack of structure and interpretability in this method compared to our model. In particular, we extract points using PIPs++ [1], a state-of-the-art point tracking method. To analyze motion, we use PCA to reduce its dimensions. We show a qualitative result in Figure 13. We observe that the motion space from point tracking is not structured nor interpretable, and thus it does not support intrepreting or manipulating motions in a meaningful way. This suggests that the unstructured point-space representation does not easily support abstracting a low-dimensional manipulatable motion representation.

### [EQJi] Failure case analysis

In Appendix D.7, we include two failure cases to analyze the boundary of our model. In particular, we find that our approach does not work with motions of shapeless objects such as fire (Figure 15), because it violates our assumption on time-invariant basis functions to represent the video appearances. Another example is chaotic unpredictable motions. We show a hair example in Figure 15 where the hair motion is unpredictable and it might not allow a frequency-based low-dimensional representation.

### [no8y] More video results on webpage

We add video visualization for motion analysis, motion smoothing, motion magnification, motion separation, and failure cases in our [supplementary webpage](https://phasepgf.github.io/more_results.html).


[1] Zheng et al., PointOdyssey: A Large-Scale Synthetic Dataset for Long-Term Point Tracking, ICCV, 2023

---

### Meta-Review · Area_Chair_CNQD · 2023-12-11

**Metareview:**

This paper presents  Phase-based neural polynomial Gabor fields for motion processing in videos, including motion magnification, loop detection, and factorization.

All reviewers rate this paper marginally above the acceptance threshold.

The initial concerns shared among the reviewers are:
- the lack of examples of complex (macro) motion,
- unclear implementation details
- lack of quantitative metrics.
- ablation study.

The authors respond to these concerns with extensive revision to clarify the implementation details and include multiple additional examples of diverse motion complexities. Overall, the reviewers agree that the authors have sufficiently addressed their concerns. The AC thus recommends to accept this paper.

**Justification For Why Not Higher Score:**

The lack of stronger support from the reviewers. Overall, the evaluation is somewhat weak due to the limited diversity of the examples. The authors gave additional results (via an external supplementary website). However, many of these results have a simple background and low resolution. While these additional examples help address partial concerns from the reviewers, the AC believes that additional validation would be needed for a higher score.

**Justification For Why Not Lower Score:**

There is a consensus among the reviewers for accepting this paper.

---

### Decision · Program_Chairs · 2024-01-16

Accept (poster)